# On the Convergence of Tsetlin Machines for the AND and the OR Operators

## Abstract

The Tsetlin Machine (TM) is an innovative machine learning algorithm rooted in propositional logic, achieving state-of-the-art performance in various pattern recognition tasks. While previous studies analyzed its convergence properties for the 1-bit and XOR operators, this work extends the analysis to the AND and OR operators, completing the study of fundamental digital operations. Our findings demonstrate that the TM almost surely converges to reproduce the AND and OR operators when trained on noise-free data over an infinite time horizon. Notably, the analysis of the OR operator uncovers a distinct property: the ability of the TM to represent two sub-patterns jointly within a single clause, contrasting with its behavior in the XOR case. Furthermore, we investigate the TM's behavior for AND/OR/XOR operators with noisy training samples, including mislabeled samples and irrelevant inputs. With wrong labels, the TM does not converge to the intended operators but can still learn efficiently. With irrelevant variables, the TM converges to the intended operators almost surely. Together, these analyses provide a comprehensive theoretical foundation for the TM's convergence properties across basic Boolean operators.

## 1 Introduction

A Tsetlin Machine (TM) (Granmo, 2018) organizes clauses, each of which is associated with a team of Tsetlin Automata (TAs) (Tsetlin, 1961), to collaboratively capture distinct sub-patterns for a certain class. A TA, which is the core learning entity of TM, is a kind of learning automata (Zhang et al., 2020; Yazidi et al., 2019; Omslandseter et al., 2022) that selects the current action based on past experiences learned from the environment in order to obtain the maximum reward. In a TM, a clause is a conjunction of literals, where a literal is a Boolean input variable or its negation. A clause is used to represent a sub-pattern. Once distinct sub-patterns are learned by a number of clauses, the overall pattern recognition task is completed by a voting scheme from the clauses. The TM has several advantages, such as transparent inference and learning (Bhattarai et al., 2024; Abeyrathna et al., 2023; Rafiev et al., 2022), and hardware friendliness (Maheshwari et al., 2023; Rahman et al., 2022; Morris et al., 2022).

The TM, together with its variations (Granmo et al., 2019; Abeyrathna et al., 2021; Darshana Abeyrathna et al., 2020; Sharma et al., 2023), has been employed in many applications, such as word sense disambiguation (Yadav et al., 2021c), aspect-based sentiment analysis (Yadav et al., 2021b), novelty detection (Bhattarai et al., 2021), text classification (Yadav et al., 2021a) with enhanced interpretability (Yadav et al., 2022), and solving contextual bandit problems (Seraj et al., 2022). These studies indicate that TMs obtain better or competitive performance compared with most of the state-of-the-art techniques. At the same time, the transparency of learning is maintained with smaller memory footprint and higher computational efficiency.

The TM convergence properties of the 1-bit operator and XOR operator were analyzed in (Zhang et al., 2022) and (Jiao et al., 2022), respectively. In (Zhang et al., 2022), TM's almost surely convergence to the identity/NOT operator with 1-bit input was confirmed, revealing the role of the hyperparameter $s$. In (Jiao et al., 2022), TM's convergence to the XOR operator with 2-bit input was proven, highlighting the functionality of the hyperparameter $T$. In this paper, we first focus on analyzing the AND and OR operators in the noise-free training samples, followed by an examina-

tion of the convergence properties of AND, OR, and XOR with noisy training samples, including the presence of wrong labels and irrelevant input variables.

This paper differs from prior work in several key aspects. While (Zhang et al., 2022) used stationary distribution analysis of discrete-time Markov chains (DTMC), the current study focuses on absorbing states. For XOR (Jiao et al., 2022), where sub-patterns are bit-wise exclusive, TM learns and converges to sub-patterns individually. In contrast, the OR operator's sub-patterns share features (e.g., $[x_1 = 1, x_2 = 1]$ and $[x_1 = 1, x_2 = 0]$ share $x_1 = 1$), allowing joint representation. We show that TM can effectively learn and represent these shared features, making the convergence process distinct. Additionally, this paper examines the role of Type II feedback, omitted in the prior work, and analyzes convergence property under noise.

It is worth noting that learning 2-bit operators, both with and without noise, is a well-solved problem with transparent solutions. Since the 1980s, numerous studies in concept learning and probably approximately correct (PAC) learning have extensively explored this topic. For instance, it has been shown in Valiant (1984); Haussler et al. (1994) that 2-DNF formulas are both properly and efficiently PAC learnable, with sample complexity scaling logarithmically in the input dimension. More generally, transparent algorithms for learning k-DNF formulas have been proposed in Marchand & Shawe-Taylor (2002), and the problem of learning conjunctions under noise has been studied in Mansour & Parnas (1998). While many elegant methods exist for learning conjunctions or disjunctions, their existence does not necessarily imply that the TM converges to such operators in the same manner. TM employs a unique approach, learning from samples to construct conjunctive expressions and coordinating these expressions across various sub-patterns, which merits its own dedicated investigation.

## 2 NOTATIONS OF THE TM

To make the article self-contained, we present the notations of TM. For more details of the inference and training concept, please refer to Appendix A.

The input of a TM is denoted as $\mathbf{X} = [x_1, x_2, \ldots, x_o]$, where $x_k \in \{0, 1\}$, $k = 1, 2, \ldots, o$, and $o$ is the number of features. A literal is either the $x_k$ in the original form or its negation $\neg x_k$. A clause is a conjunction of literals, and each literal is associated with a TA. The TA is a 2-action learning automaton whose job is to decide whether to Include/Exclude its literal in/from the clause, and the decision is determined by the current state of the TA. A clause is associated with $2o$ TAs, forming a TA team. A TA team is denoted in general as $\mathcal{G}_j^i = \{\text{TA}_{k'}^{i,j} | 1 \leq k' \leq 2o\}$, where $k'$ is the index of the TA, $j$ is the index of the TA team/clause (multiple TA teams form a TM), and $i$ is the index of the TM/class to be identified (Here a TM identifies a class, multiple TMs identify multiple classes).

Suppose we are investigating the $i^{th}$ TM whose job is to identify class $i$, and that the TM is composed of $m$ TA teams. Then $C_j^i(\mathbf{X})$ can be used to denote the output of the $j^{th}$ TA team, which is a conjunctive clause:

$$\text{For training}: C_j^i(\mathbf{X}) = \begin{cases} \left( \bigwedge_{k \in \xi_j^i} x_k \right) \wedge \left( \bigwedge_{k \in \bar{\xi}_j^i} \neg x_k \right), & \text{for } \xi_j^i,\ \bar{\xi}_j^i \neq \emptyset, \\ 1, & \text{for } \xi_j^i,\ \bar{\xi}_j^i = \emptyset. \end{cases} \tag{1}$$

$$\text{For testing}: C_j^i(\mathbf{X}) = \begin{cases} \left( \bigwedge_{k \in \xi_j^i} x_k \right) \wedge \left( \bigwedge_{k \in \bar{\xi}_j^i} \neg x_k \right), & \text{for } \xi_j^i,\ \bar{\xi}_j^i \neq \emptyset, \\ 0, & \text{for } \xi_j^i,\ \bar{\xi}_j^i = \emptyset. \end{cases} \tag{2}$$

In Eqs. (1) and (2), $\xi_j^i$ and $\bar{\xi}_j^i$ are defined as the sets of indexes for the literals that have been included in the clause. $\xi_j^i$ contains the indexes of included original (non-negated) features, $x_k$, whereas $\bar{\xi}_j^i$ contains the indexes of included negated features, $\neg x_k$.

Each clause represents a sub-pattern associated with class $i$ by including a literal (a feature or its negation) if it contributes to the sub-pattern, or excluding it when deemed irrelevant. Multiple

clauses, i.e., the TA teams, are assembled into a complete TM to sum up the outputs of the clauses $f_{\sum}(\mathcal{C}^i(\mathbf{X})) = \sum_{j=1}^{m} C_j^i(\mathbf{X})$, where $\mathcal{C}^i(\mathbf{X})$ is the set of clauses for class $i$. The output of the TM is further determined by the unit step function: $\hat{y}^i = \begin{cases} 0, & \text{for } f_{\sum}(\mathcal{C}^i(\mathbf{X})) < Th \\ 1, & \text{for } f_{\sum}(\mathcal{C}^i(\mathbf{X})) \geq Th \end{cases}$, where $Th$ is a predefined threshold for classification. This is indeed a voting scheme. For example, the classifier $(x_1 \wedge \neg x_2) + (\neg x_1 \wedge x_2)$ captures the XOR-relation when $Th = 1$, meaning if any sub-pattern is satisfied, the input will be identified as following the XOR logic.

Note that the TM can assign a polarity to each TA team (Granmo, 2018), and one can refer to Appendix A for more information. In this study, for ease of analysis, we consider only positive polarity clauses. Nevertheless, this does not change the nature of TM learning.

For training, the labeled data $(\mathbf{X} = [x_1, x_2, ..., x_o], y^i)$ is given to TM, and each TA is guided by Type I and Type II Feedback defined in Tables 1 and 2, respectively. Type I Feedback is triggered when the training sample has a positive label: $y^i = 1$, while Type II feedback is utilized when $y^i = 0$. The parameter, $s$, controls the granularity of the clauses. NA in these tables means not applicable. Examples demonstrating TA state transitions per feedback tables can be found in Section 3.1 in (Zhang et al., 2022). In brief, Type I feedback is to reinforce true positive and Type II feedback is to fight against false negative.

| Value of the clause $C_j^i(\mathbf{X})$ | | 1 | | 0 | |
| --- | --- | --- | --- | --- | --- |
| Value of the Literal $x_k/\neg x_k$ | | 1 | 0 | 1 | 0 |
| **Include Literal** | $P(\text{Reward})$ | $\frac{s-1}{s}$ | NA | 0 | 0 |
| | $P(\text{Inaction})$ | $\frac{1}{s}$ | NA | $\frac{s-1}{s}$ | $\frac{s-1}{s}$ |
| | $P(\text{Penalty})$ | 0 | NA | $\frac{1}{s}$ | $\frac{1}{s}$ |
| **Exclude Literal** | $P(\text{Reward})$ | 0 | $\frac{1}{s}$ | $\frac{1}{s}$ | $\frac{1}{s}$ |
| | $P(\text{Inaction})$ | $\frac{1}{s}$ | $\frac{s-1}{s}$ | $\frac{s-1}{s}$ | $\frac{s-1}{s}$ |
| | $P(\text{Penalty})$ | $\frac{s-1}{s}$ | 0 | 0 | 0 |

Table 1: Type I Feedback — Feedback upon receiving a sample with label $y^i = 1$ (Granmo, 2018).

| Value of the clause $C_j^i(\mathbf{X})$ | | 1 | | 0 | |
| --- | --- | --- | --- | --- | --- |
| Value of the Literal $x_k/\neg x_k$ | | 1 | 0 | 1 | 0 |
| **Include Literal** | $P(\text{Reward})$ | 0 | NA | 0 | 0 |
| | $P(\text{Inaction})$ | 1.0 | NA | 1.0 | 1.0 |
| | $P(\text{Penalty})$ | 0 | NA | 0 | 0 |
| **Exclude Literal** | $P(\text{Reward})$ | 0 | 0 | 0 | 0 |
| | $P(\text{Inaction})$ | 1.0 | 0 | 1.0 | 1.0 |
| | $P(\text{Penalty})$ | 0 | 1.0 | 0 | 0 |

Table 2: Type II Feedback — Feedback upon receiving a sample with label $y^i = 0$ (Granmo, 2018).

To avoid the situation that a majority of the TA teams learn only a subset of sub-patterns, forming an incomplete representation[1], the hyperparameter $T$ is used to regulate the resource allocation. The strategy works as follows (Granmo, 2018):

**Generating Type I Feedback.** If the label of the training sample $\mathbf{X}$ is $y^i = 1$, we generate, in probability, *Type I Feedback* for each clause $C_j^i \in \mathcal{C}^i$ according to:

$$u_1 = \frac{T - \max(-T, \min(T, f_{\sum}(\mathcal{C}^i(\mathbf{X}))))}{2T}. \tag{3}$$

---

[1] For example, for the XOR operator, one should avoid the situation that a majority of TA teams converge to $\neg x_1 \wedge x_2$ to represent the sub-pattern of $[0, \ 1]$, and ignore the other sub-pattern $[1, \ 0]$.

**Generating Type II Feedback.** If the label of the training sample $\mathbf{X}$ is $y^i = 0$, we generate, again, in probability, *Type II Feedback* to each clause $C_j^i \in \mathcal{C}^i$ according to:

$$u_2 = \frac{T + \max(-T, \min(T, f_{\sum}(\mathcal{C}^i(\mathbf{X}))))}{2T}. \tag{4}$$

Briefly speaking, when the number of clauses representing one sub-pattern reaches $T$, learning from that sub-pattern will stop as the probability of triggering update is 0 according to Eq. (3) for positive polarity. The same concept applies according to Eq. (4) for negative polarity.

## 3    CONVERGENCE ANALYSIS OF THE AND OPERATOR

A TM has converged when the transitions among the states of its TAs do not happen any longer. We assume that the training samples are noise free, i.e., $P(y = 1|x_1 = 1, x_2 = 1) = 1, P(y = 0|x_1 = 0, x_2 = 1) = 1, P(y = 0|x_1 = 1, x_2 = 0) = 1, P(y = 0|x_1 = 0, x_2 = 0) = 1$. We also assume the training samples are independently drawn at random, and the above four cases will appear with non-zero probability, which means that all of the four types of samples will appear for infinite number of times given infinite time horizon.

Because the considered AND operator has only one pattern of input, i.e., $x_1 = 1, x_2 = 1$, that will trigger a true output, we employ one clause in this TM, and we thus can ignore the indices of the classes and the clauses in our notation in the proof. After simplification, $\text{TA}_k^{i,j}$ becomes $\text{TA}_k$, and $C_1^1$ becomes $C$. Since there are two input variables, namely $x_1$ and $x_2$, we implement four TAs in the clause, i.e., $\text{TA}_1, \text{TA}_2, \text{TA}_3$, and $\text{TA}_4$. $\text{TA}_1$ has two actions, i.e., including or excluding $x_1$. Similarly, $\text{TA}_2$ corresponds to including or excluding $\neg x_1$. $\text{TA}_3$ and $\text{TA}_4$ determine the behavior of $x_2$ and $\neg x_2$, respectively. Once the TM can converge correctly to the intended operation, the actions of $\text{TA}_1, \text{TA}_2, \text{TA}_3$, and $\text{TA}_4$ should be I, E, I, and E. Here we use "I" and "E" as abbreviations for include and exclude respectively.

**Theorem 1.** *Any clause will converge almost surely to $x_1 \wedge x_2$ given noise free AND training samples in infinite time when $u_1 > 0$ and $u_2 > 0$.*

Due to page limit, the complete proof of Theorem 1 can be found in Appendix B. We here outline the concept and the main steps of the proof.

The condition $u_1 > 0$ and $u_2 > 0$ guarantees that all types of samples are always given and no specific type is blocked by Eqs. (3) and (4) during training. The goal of the proof is to show that the system transitions will guarantee that there is a unique absorbing state of the TM and the absorbing state has the actions of $\text{TA}_1, \text{TA}_2, \text{TA}_3$, and $\text{TA}_4$ to be I, E, I, E, respectively, corresponding to the propositional expression $x_1 \wedge x_2$.

To simplify the complex analysis of joint TA transitions, we use quasi-stationary analysis by freezing the transitions of the TAs for the first input bit and focusing on the transitions of the second input bit. Clearly, there are four possibilities for the first bit $x_1$. We name them as cases, as: **Case 1:** $\text{TA}_1 = \text{E}, \text{TA}_2 = \text{I}$, i.e., include $\neg x_1$. **Case 2:** $\text{TA}_1 = \text{I}, \text{TA}_2 = \text{E}$, i.e., include $x_1$. **Case 3:** $\text{TA}_1 = \text{E}, \text{TA}_2 = \text{E}$, i.e., exclude both $x_1$ and $\neg x_1$. **Case 4:** $\text{TA}_1 = \text{I}, \text{TA}_2 = \text{I}$, i.e., include both $x_1$ and $\neg x_1$.

In each of the above four cases, we analyze individually the transition of $\text{TA}_3$ with a given current action, for different actions of $\text{TA}_4$, and vice versa. We index the possibilities as situations: **Situation 1.** We study the transition of $\text{TA}_3$ when it has "Include" as its current action, given different actions of $\text{TA}_4$ (i.e., when the action of $\text{TA}_4$ is frozen as "Include" or "Exclude"). **Situation 2.** We study the transition of $\text{TA}_3$ when it has "Exclude" as its current action, given different actions of $\text{TA}_4$ (i.e., when the action of $\text{TA}_4$ is frozen as "Include" or "Exclude"). **Situation 3.** We study the transition of $\text{TA}_4$ when it has "Include" as its current action, given different actions of $\text{TA}_3$ (i.e., when the action of $\text{TA}_3$ is frozen as "Include" or "Exclude"). **Situation 4.** We study the transition of $\text{TA}_4$ when it has "Exclude" as its current action, given different actions of $\text{TA}_3$ (i.e., when the action of $\text{TA}_3$ is frozen as "Include" or "Exclude").

Within each of the situation, there are 8 possible instances, determined by 4 possible combinations of the input variables of $x_1$ and $x_2$, and the two possible TA actions, Include and Exclude. As an example, we randomly select an instance in Case 1, Situation 1. The selected instance is when the

training sample is ($[x_1 = 1, x_2 = 1]$, $y = 1$), and TA$_4$ is E. For this instance, the training sample will trigger Type I feedback because $y = 1$. Based on the current status of the TAs, the clause is in the form $C = \neg x_1 \wedge x_2$, with value 0. In Situation 1, the studied TA is TA$_3$, its corresponding literal is thus $x_2$, with value 1. Given $y = 1$, clause value 0, literal value 1, we go to Table 1, the third column of transition probabilities for "Include Literal", and find the transition of TA$_3$ to be: the penalty probability $\frac{1}{s}$ and the inaction probability $\frac{s-1}{s}$. To indicate the transitions of TA$_3$, we have plotted the transition diagram in Fig. 1, where $P$ and $R$ represent Reward and Penalty respectively. Note that the overall transition probability is $u_1 \frac{1}{s}$, where $u_1$ is defined in Eq. (3). Here, we have assumed $u_1 > 0$.

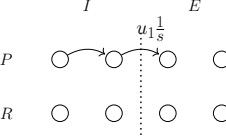

Figure 1: Transition of TA$_3$ when its current action is Include, TA$_1$, TA$_2$, and TA$_4$'s actions are Exclude, Include, and Exclude, respectively, upon a training sample ($x_1 = 1$, $x_2 = 1$, $y = 1$).

To complete the quasi-stationary analysis of TA$_3$ and TA$_4$, we must in total analyze $4 \times 4 \times 8 = 128$ transition instances, similar to the diagram in Fig. 1.

Based on the analysis of the 128 transition instances, we can summarize the transitions of TA$_3$ and TA$_4$. By observing the transition directions, we can conclude that there is a unique absorbing state for TA$_3$ and TA$_4$, given TA$_1$ and TA$_2$ being frozen as I, and E respectively. The absorbing state is when TA$_3$ and TA$_4$ are in I and E respectively. Once this step is completed, we must freeze TA$_3$ and TA$_4$, and study the transitions of TA$_1$ and TA$_2$ in the same way. Thereafter, we can conclude that the system has a unique absorbing state, which is TA$_1$, TA$_2$, TA$_3$, and TA$_4$ being in I, E, I, E respectively, in the full dynamics of the system.

## 4 CONVERGENCE ANALYSIS OF THE OR OPERATOR

We assume the training samples for the OR operator are noise free (i.e., Eq. (5)), and are independently drawn at random. All those four cases will appear with non-zero probability.

$$P(y = 1 | x_1 = 1, x_2 = 1) = 1, \ P(y = 1 | x_1 = 0, x_2 = 1) = 1, \tag{5}$$
$$P(y = 1 | x_1 = 1, x_2 = 0) = 1, \ P(y = 0 | x_1 = 0, x_2 = 0) = 1.$$

**Theorem 2.** *The clauses in a TM can almost surely learn the 2-bit OR logic given noise free training samples (shown in Eq. (5)) in infinite time, when $T \leq \lfloor \frac{m}{2} \rfloor$.*

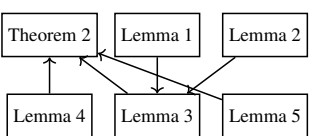

Figure 2: The dependence for the proof of the Theorem 2.

The proof of the theorem requires Lemma 1-Lemma 5 and their dependence is shown in Fig. 2. Clearly, there are three sub-patterns for the OR operator. In Lemma 1, we will show that any clause is able to converge to an intended sub-pattern when the training sample of only one sub-pattern is given, and when $u_1 > 0$ and $u_2 > 0$. In Lemma 2, we will show that the TM will become recurrent (not absorbing) when more sub-patterns jointly appear in the training samples and when $u_1 > 0$ and $u_2 > 0$. These two lemmas will be utilized in the proof of Lemma 3. Lemma 2 also reveals the recurrent nature of TM for the OR operator when the functionality of $T$ is not enabled, i.e., when $u_1 > 0$ and $u_2 > 0$. This confirms the necessity of enabling the functionality of $T$ in order to converge to an absorbing state that fulfills the OR operator, to be indicated by Lemma 3-Lemma 5. Specifically, Lemma 3-Lemma 5 analyze the system behavior when $T$ is enabled and how $T$ should

be configured for the TM to converge to the OR operator. They guarantee that when the system arrives an absorbing state, any sample from the intended sub-patterns will offer a vote sum no less than $T$ while the sample from the unintended sub-pattern has a vote sum 0. Then the OR operator can be inferred by setting $Th = T$. In what follows, we will present and prove the lemmas.

**Lemma 1.** *For any one of the three sub-patterns of $x_1$ and $x_2$ resulting in $y = 1$, shown in Eqs. (6)-(8), the TM can converge to the intended sub-pattern when noise free training samples following this sub-pattern are given, and when $u_1 > 0$, $u_2 > 0$.*

$$P(y = 1|x_1 = 1, x_2 = 1) = 1, \ P(y = 0|x_1 = 0, x_2 = 0) = 1, \tag{6}$$

$$P(y = 1|x_1 = 0, x_2 = 1) = 1, \ P(y = 0|x_1 = 0, x_2 = 0) = 1, \tag{7}$$

$$P(y = 1|x_1 = 1, x_2 = 0) = 1, \ P(y = 0|x_1 = 0, x_2 = 0) = 1. \tag{8}$$

The proof of Lemma 1 involves demonstrating convergence for three sub-patterns: those governed by Eqs. (6), (7), and (8). These analyses build upon the convergence proofs for the XOR and AND operators. For the sub-pattern in Eq. (6), transition diagrams in Appendix B confirm that the TAs converge to $TA_1 = I$, $TA_2 = E$, $TA_3 = I$, and $TA_4 = E$, when input samples $[\boldsymbol{x}_1 = 0, \boldsymbol{x}_2 = 1]$ and $[\boldsymbol{x}_1 = 1, \boldsymbol{x}_2 = 0]$ are excluded. The other two sub-patterns are proven using similar principles. Full details are provided in Appendix C.

From Lemma 1, we show that the clauses converge to the intended sub-pattern if the training samples following this particular sub-pattern are given. From Lemma 2, we will show that the system becomes recurrent if any two or more sub-patterns of training samples are given. Specifically, we show the TM is recurrent given samples following Eq. (5) and Eqs. (9)-(11), when $u_1 > 0$, $u_2 > 0$.

$$P(y = 1|x_1 = 1, x_2 = 1) = P(y = 1|x_1 = 1, x_2 = 0) = P(y = 0|x_1 = 0, x_2 = 0) = 1, \tag{9}$$

$$P(y = 1|x_1 = 1, x_2 = 1) = P(y = 1|x_1 = 0, x_2 = 1) = P(y = 0|x_1 = 0, x_2 = 0) = 1, \tag{10}$$

$$P(y = 1|x_1 = 1, x_2 = 0) = P(y = 1|x_1 = 0, x_2 = 1) = P(y = 0|x_1 = 0, x_2 = 0) = 1. \tag{11}$$

**Lemma 2.** *The TM becomes recurrent if any two or more of the three sub-patterns jointly appear in the training samples, as shown in Eqs. (5), (9)-(11), when $u_1 > 0$, $u_2 > 0$.*

**Proof of Lemma 2:** To show the recurrent property when samples following Eq. (9) are given, we need to show that the absorbing states for Eq. (6) disappear when $([x_1 = 1, x_2 = 0], y = 1)$ is given in addition, and the same applies for Eq. (8) when $([x_1 = 1, x_2 = 1], y = 1)$ is given.

We first show that the absorbing state of $TA_1 = I$, $TA_2 = E$, $TA_3 = I$, $TA_4 = E$, for sub-pattern $([x_1 = 1, x_2 = 1], y = 1)$ as shown in Eq. (6), disappears when sub-pattern $([x_1 = 1, x_2 = 0], y = 1)$ is given in addition. Indeed, $TA_3$ will move toward E when $([x_1 = 1, x_2 = 0], y = 1)$ is given, because a penalty is given to $TA_3$ as shown in Fig. 3.

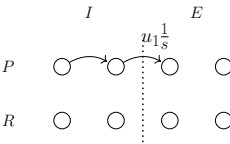

Figure 3: Transition of $TA_3$ when its current action is Include, $TA_1$, $TA_2$, and $TA_4$'s actions are Include, Exclude, and Exclude, respectively, upon a training sample $(x_1 = 1, x_2 = 0, y = 1)$.

Clearly, when $([x_1 = 1, x_2 = 0], y = 1)$ is given in addition, $TA_3$ has a non-zero probability to move towards "Exclude". Therefore, "Include" is not the only direction that $TA_3$ moves to upon the new input. In other words, $([x_1 = 1, x_2 = 0], y = 1)$ will make the state $TA_1 = I$, $TA_2 = E$, $TA_3 = I$, $TA_4 = E$, not absorbing any longer. For other states, the newly added training sample will not remove any transition from the previous case. For this reason, the system will not have any new absorbing state. Therefore, when $([x_1 = 1, x_2 = 0], y = 1)$ is given in addition, the absorbing state disappears and the system will not have any new absorbing state.

Following the same concept, we show that the absorbing state for $([x_1 = 1, x_2 = 0], y = 1)$ shown in Eq. (8), i.e., $TA_1 = I$, $TA_2 = E$, $TA_3 = E$, $TA_4 = I$, disappears when sub-pattern

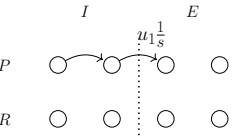

Figure 4: Transition of $TA_4$ when its current action is Include, $TA_1$, $TA_2$, and $TA_3$'s actions are Include, Exclude, and Exclude, respectively, upon a training sample ($x_1 = 1$, $x_2 = 1$, $y = 1$).

($[x_1 = 1, x_2 = 1], y = 1$) is given in addition. Indeed, $TA_4$ will also move towards E when ($[x_1 = 1, x_2 = 1], y = 1$) is given, as shown in Fig. 4.

Understandably, because of the newly added sub-patterns, the absorbing states in Eqs. (6) and (8) disappear and no new absorbing states are generated. In other words, the TM trained based on samples from Eq. (9) becomes recurrent.

Following the same concept, we can show that the system becomes recurrent for Eqs. (5), (10), and (11) as well. For the sake of conciseness, we will not provide the details here. In general, any newly added sub-pattern will involve a probability for the learnt sub-pattern to move outside the learnt state, making the system recurrent. ∎

Lemma 2 tells us that if we always give TM the training samples from all sub-patterns without blocking the learnt patterns by using $T$ via Eqs. (3) and (4), the system is recurrent. In other words, if we want to have the TM converge to the OR operator in an absorbing state, it is critical to utilize the feature of $T$ to block any incoming training samples from updating the learnt sub-patterns. Specifically, we need to configure $T$ (1) so that the absorbing states exist and (2) confirm that the absorbing states follows the OR operator. In what follows, we will, through Lemmas 3-5, show how $T$ via Eqs. (3) and (4) can guarantee the convergence and how the value of $T$ should be configured.

Let's revisit the functionality of $T$. $T$ can block the training samples from updating a learnt sub-pattern (clauses that have converged to one of the absorbing states) so that the clauses that have not converged can be guided to learn the other unlearned sub-patterns. More specifically, if the vote sum of the clauses reaches $T$ for a certain sub-pattern, the new training samples of this sub-pattern will be blocked by the TM. There are three sub-patterns in OR operator. When the sum of clauses for each of the three sub-patterns reaches $T$, all training samples for Type I feedback are blocked. At the same time, if all samples for Type II feedback will not trigger any update to the states of TAs, the TM is absorbed. In Lemma 3, we detail the necessity and sufficiency of the absorbing state.

**Lemma 3.** *The system is absorbed if and only if (1) the vote sum of any sample from intended sub-patterns reaches $T$, i.e., $f_\Sigma(C_i(\mathbf{X})) = T$, $\forall \mathbf{X} = [x_1 = 0, x_2 = 1]$ or $[x_1 = 0, x_2 = 1]$ or $[x_1 = 0, x_2 = 1]$, and (2) no clause is formed only by a negated literal or negated literals.*

**Proof of Lemma 3:** In Lemma 2, the TM is recurrent if the functionality of $T$ is disabled (i.e., $u_1 > 0$, $u_2 > 0$). Therefore, for the OR operator to converge, the functionality of $T$ is critical to block any feedback in order to form an absorbing state.

By design, TM will either be updated via Type I feedback or Type II feedback. We show via (1) the condition when Type I feedback is blocked and then show via (2) when any update from Type II feedback is not given. When both types of feedback are blocked, the system will not be updated anymore and thus absorbed.

To prove (1) in Lemma 3, we show that the system is not absorbed when 0 or 1 intended sub-pattern is blocked by $T$. When 2 intended sub-patterns are blocked, the system will guide the clauses to learning the remaining intended sub-pattern. Only when all 3 intended sub-patterns are blocked by $T$, the system will stop updating based on Type I feedback.

Clearly, when no intended sub-pattern is blocked by $T$, the training samples given to the system follow Eq. (5). Following this type of training samples, it has already been shown in Lemma 2 that the TM is recurrent. When only 1 intended sub-pattern is blocked by $T$, the system is updated based on Eqs. (9), (10), or (11), which is also recurrent.

We look at the cases when two intended sub-patterns are blocked by $T$ but the third is not blocked. In other words, the vote sum for any two intended sub-patterns reaches at least $T$, and the sum for the remaining sub-pattern is less than $T$. In this case, only one type of the samples from Eqs. (6) or (7) or (8) will be given to the TM. Based on Lemma 1, we understand that all clauses, including the ones that follow the two blocked sub-patterns, will be reinforced to learn the unblocked sub-pattern. This is due to the fact that only the samples following the unblocked sub-pattern are given to the TMs. In this circumstance, as soon as the unblocked sub-pattern also has $T$ clauses, i.e., when all three sub-patterns are blocked by $T$ at the same time, Type I feedback will be blocked completely.

Note that the samples from the unblocked sub-pattern will encourage the learnt clauses (the clauses that follow sub-patterns with vote sum $T$) move out from the learnt sub-patterns, making the vote sum of learnt sub-patterns being lower than $T$ again. If this happens before the vote sum of the to-be-learnt sub-pattern reaches $T$, two sub-patterns will be unblocked and the system becomes one of three cases described by Eqs. (9), (10) or (11). In other words, even if an absorbing state exists when two intended sub-patterns are already blocked by $T$, the system may not monotonically move towards the absorbing state. Nevertheless, as soon as all three intended sub-patterns are blocked by reaching $T$, the Type I feedback will be blocked.

Here we prove (2) in Lemma 3. Type II feedback is only triggered by training sample ($[x_1 = 0, x_2 = 0]$, $y = 0$) in the OR operator. For Type II feedback, based on Table 2, any transition is only triggered as a penalty when excluded literal has 0 value and the clause is evaluated as 1. Specifically for the OR operation, this only happens when $C = \neg x_1 \wedge \neg x_2$ or $C = \neg x_1$ or $C = \neg x_2$. For $C = \neg x_1 \wedge \neg x_2$, based on the Type II feedback, the TA with the action "excluding $x_1$" and the TA with the action "excluding $x_2$" will be penalized. In other words, the actions of the TAs for $x_1$ and $x_2$ will be encouraged to move from exclude to include side. As soon as any one of TAs for $x_1$ or $x_2$ (or occasionally both of them) becomes included, the clause will become $C = x_1 \wedge \neg x_1 \wedge \neg x_2$ or $C = \neg x_1 \wedge x_2 \wedge \neg x_2$ (or occasionally $C = x_1 \wedge \neg x_1 \wedge x_2 \wedge \neg x_2$). In this case, input $[x_1 = 0, x_2 = 0]$ will always result in 0 as the output of the clause and then the Type II feedback will not update the system any longer. Following the same concept, for $C = \neg x_2$, the Type II feedback will encourage the excluded $x_1$ to be included so that the clause becomes $C = x_1 \wedge \neg x_2$. The same applies to $C = \neg x_1$, which will eventually become $C = \neg x_1 \wedge x_2$ upon Type II feedback. When all clauses in $C = \neg x_2$ or $C = \neg x_1$ are also updated to $C = x_1 \wedge \neg x_2$ or $C = \neg x_1 \wedge x_2$, no Type II feedback is triggered up on any input sample.

We summarize the requirements for an absorbing state:

- For any sample following $\mathbf{X} = [x_1 = 1, x_2 = 1]$, or $\mathbf{X} = [x_1 = 1, x_2 = 0]$, or $\mathbf{X} = [x_1 = 0, x_2 = 1]$, the vote sum of clauses, i.e., $f_\Sigma(C^i(\mathbf{X}))$ must be at least $T$, no matter in which form the clauses are constructed. This will block any Type I feedback.

- There are no clauses with literal(s) in only negated form, such as $C = \neg x_1$ or $C = \neg x_2$ or or $C = \neg x_1 \wedge \neg x_2$. This guarantees no transition happens upon any Type II feedback. ∎

In Lemma 3, we find the conditions of the absorbing state. In the next Lemma, we will show how to set up the value of $T$ so that the vote sum for each intended sub-pattern can indeed reach $T$.

**Lemma 4.** $T \leq \lfloor m/2 \rfloor$ *is required so that the vote sum of any sample from intended sub-patterns can reach $T$.*

**Proof of Lemma 4:** There are three intended sub-patterns in the OR operator. Given $m$ clauses in total, to make sure each one has at least $T$ votes, we have $3T \leq m$. This requires $T \leq \lfloor m/3 \rfloor$ for any integer. However, the nature of the OR operator offers the possibility to represent 2 sub-patterns jointly. For example, $T$ clauses in the form of $x_1$ will result in the vote sum as $T$ for both $[x_1 = 1, x_2 = 0]$ and $[x_1 = 1, x_2 = 1]$. If there are other $T$ clauses to represent the remaining sub-pattern, in total $2T$ clauses can offer the vote sum as $T$ for all intended sub-patterns. We thus have $2T \leq m$, giving $T \leq \lfloor m/2 \rfloor$ for any integer. Note that the fact that two sub-patterns can be jointly represented has been observed and confirmed in experiments shown in Appendix F.

When we have a smaller $T$, different sub-patterns may be represented by distinct clauses, offering more flexibility. However, when $T > \lfloor m/2 \rfloor$, there will always be one or two sub-patterns that cannot obtain a sum of $T$ clauses. For this reason, the maximum integer value is $T = \lfloor m/2 \rfloor$. ∎

In Lemma 5, we show that the input sample $[x_1 = 0, x_2 = 0]$ will not give a vote sum greater than or equal to $T$. This is to avoid any possible false positive upon input $[x_1 = 0, x_2 = 0]$ in testing.

**Lemma 5.** *When absorbing, the sample from unintended sub-pattern, i.e., $[x_1 = 0, x_2 = 0]$, will not give any vote sum greater than or equal to $T$.*

**Proof of Lemma 5:** Obviously, to have a positive output form $[x_1 = 0, x_2 = 0]$, the clause should be in the form of $C = \neg x_1$ or $C = \neg x_2$ or $C = \neg x_1 \wedge \neg x_2$. It has already shown in the proof of Lemma 3 that Type II feedback will eliminate such clauses. In fact, when the system is absorbed, no clause will be in the form of $C = \neg x_1$ or $C = \neg x_2$ or $C = \neg x_1 \wedge \neg x_2$. For this reason, $[x_1 = 0, x_2 = 0]$ will never result in a sum of clause outputs greater than or equal to $T$. ∎

**Proof of Theorem 2:** Based on Lemma 3–Lemma 5, we understand that if $T \leq \lfloor m/2 \rfloor$ holds, Type I feedback will eventually be blocked and Type II feedback will eventually only give "inaction" feedback. In this situation, no actual transition will be triggered and thus the system reaches the absorbing state. Before absorbed, the system moves back and forth in the intermediate states. Once absorbed, samples from any one of the intended sub-patterns will result in a vote sum to no less than $T$ and the unintended sub-pattern will have a vote sum to 0. We thus have the OR logic almost surely by setting a threshold $Th = T$ and conclude the proof. ∎

Now let's study a simple example with $m = 2$, $T = 1$. Here, $C_1 = x_1$ and $C_2 = x_2$ can be an instance for an absorbing case. $C_1 = x_1$ and $C_2 = \neg x_1 \wedge x_2$ also works. Clearly, the clauses can be in various forms, as long as the conditions in Lemma 3 fulfill. These converged clauses are not necessarily in the exact form of the three sub-patterns, which is distinct to that of the XOR operator.

**Remark 1.** *Although both AND and OR operators converge, the approaches are different. For AND operator, the system is converged because the clauses become eventually absorbed to the intended pattern upon Type I and Type II feedback, even if the functionality of $T$ is disabled ($u_1 > 0$ and $u_2 > 0$). As the TM enables the functionality of $T$ by default, the system will be absorbed when $T$ clauses converge to $x_1 \wedge x_2$, before all clauses converge to this pattern. However, for the OR operator, the functionality of $T$ is critical because the TM is recurrent if $u_1 > 0$ and $u_2 > 0$. The absorbing state of the OR operator is achieved because the functionality of $T$ blocks all Type I feedback and Type II feedback gives only "Inaction" feedback. The concept of convergence for the OR operator is similar to that of XOR, but the form of clauses after absorbing varies due to the possible joint representation of sub-patterns in OR.*

**Remark 2.** *When $T$ is greater than half of the number of the clauses, i.e., $T > \lfloor m/2 \rfloor$, the system will not have an absorbing state. We conjecture that the system can still learn the sub-patterns in an unbalanced manner, as long as $T$ is not configured too close to the total number of clauses $m$.*

Given $T > \lfloor m/2 \rfloor$, Type I feedback cannot be completely blocked and the TM is recurrent. Nevertheless, if $T$ is not close to $m$, there will be clauses that possibly learn distinct sub-patterns. In addition, Type II feedback can avoid the form of $C = \neg x_1$ or $C = \neg x_2$ or $C = \neg x_1 \wedge \neg x_2$ from happening. Therefore, with $Th > 0$, the TM may still learn the OR operator with high probability.

To validate the theoretical analyses, we present in Appendix F the experiment results[2] for both the AND and the OR operators, confirming the correctness of the above theorems.

# 5 REVISIT THE XOR OPERATOR

Let us revisit the proof of XOR operator. As stated in (Jiao et al., 2022), when the system is absorbed, the clauses follow the format $C = x_1 \wedge \neg x_2$ or $C = \neg x_1 \wedge x_2$ precisely. In other words, a clause with just one literal, such as $C = x_1$, cannot absorb the system. The main reason is that the sub-patterns in XOR operator are mutual exclusive, i.e., the sub-patterns cannot be merged in any way. Although Type I feedback can be blocked when $T$ clauses follow one sub-pattern using one literal, the Type II feedback can reinforce the other missing literal to be included. For example, when $T$ clauses happens to converge to $C = x_1$, the Type I feedback from any input samples of $([x_1 = 1, x_2 = 0], y = 1)$ will be blocked. In this situation, the unblocked Type II feedback from $([x_1 = 1, x_2 = 1], y = 0)$ will encourage the clause to include $\neg x_2$. This is because upon a sample

---

[2]The code for the experiments of this paper can be found at https://github.com/JaneGlim/Convergence-of-Tsetline-Machine-for-the-AND-OR-operators.

$([x_1 = 1, x_2 = 1], y = 0)$, we have Type II feedback, $C = x_1 = 1$, and the studied literal is $\neg x_2 = 0$. When the TA for excluding $\neg x_2$ is considered, a big penalty, i.e., 1, is given to the TA, making it moving towards action *Included*, and thus $C = x_1$ eventually becomes $C = x_1 \wedge \neg x_2$. Following the same concept, we can analyze the development for $C = \neg x_1$, $C = x_2$, and $C = \neg x_2$, which will eventually converge to $C = \neg x_1 \wedge x_2$ or $C = x_1 \wedge \neg x_2$, upon Type II feedback.

## 6 Convergence Analysis under Random Noise

We studied the convergence properties of AND, OR, and XOR operators under training samples with noise following the noise type named *noisy completely at random* Frénay & Verleysen (2013), categorized as wrong labels (in Appendix D) and irrelevant input variables (in Appendix E). A wrong label refers to an input that should be labeled as 1 but is instead labeled as 0, or vice versa. An irrelevant input variable, on the other hand, is one that does not contribute to the classification. We demonstrate that, with wrong labels, the TM does not converge to the intended operators but can still learn efficiently. With irrelevant variables, the TM converges to the intended operators almost surely. Experimental results confirmed these findings (Appendix G). We summarize the main findings below. The proof and the experiment results can be found in the corresponding appendices.

**Theorem 3.** *The TM is recurrent given training samples with wrong labels for the AND, OR, and XOR operators.*

**Remark 3.** *The recurrent property of TM indicates that there is a non-zero probability that it cannot learn the intended operator. The primary reason for the recurrent behavior when wrong labels are present is the statistically conflicting labels for the same input samples. These inconsistency causes the TAs within a clause to learn conflicting outcomes for the same input. When a clause learns to evaluate an input as 1 based on Type I feedback, samples with a label of 0 for the same input prompt it to learn the input as 0 during Type II feedback. This conflict in labels confuses the TM, leading to back-and-forth learning.*

**Remark 4.** *Note that although wrong labels will make the TM not converge (not absorbing with 100% accuracy for the intended logic), via experiments, we can still find that the TM are able to learn the operators efficiently, as shown in Appendix G. This property aligns with the concept of PAC learnable Mansour & Parnas (1998) or $\epsilon$-optimality Zhang et al. (2020), although a formal proof remains an open question.*

**Theorem 4.** *The clauses in a TM can almost surely learn the 2-bit AND logic given training samples with $k$ irrelevant input variables in infinite time, $0 < k < \infty$, when $T \leq m$.*

**Theorem 5.** *The clauses in a TM can almost surely learn the 2-bit XOR and OR logic given training samples with $k$ irrelevant input variables in infinite time, $0 < k < \infty$, when $T \leq \lfloor m/2 \rfloor$.*

When the number of irrelevant variables is large, the training set may not cover all possible examples due to the required exponential space. Although not yet theoretically proven, polynomial space for training samples seems feasible for TM, which has been observed by experiments (Appendix G.3). This is because the TM can independently update the actions of a TA within a clause, as long as the clause value and the literal value are determined by the training sample. In other words, once the clause value and the literal value are known, the transitions triggered by Type I and Type II feedback are fully determined. As a result, the TM does not need to observe all possible combinations of irrelevant input variables to learn effectively. Instead, as long as the statistical irrelevance of certain variables is demonstrated in the training samples, the corresponding TA transitions will be triggered accordingly. This enables the TM to learn without requiring exhaustive coverage of the input space.

## 7 Conclusions

In this article, we prove the convergence of the TM for the AND and OR operators with noise free training samples. Our proof for the OR operator highlights the TM's ability to learn joint sub-patterns, showcasing the efficiency of its learning process. Additionally, we analyze the behavior of the TM for the AND, OR, XOR operators in the presence of random noise within the training data. Combined with the convergence proofs in (Zhang et al., 2022) and (Jiao et al., 2022), this work concludes the analysis of TM convergence for fundamental digital operators.

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

## A APPENDIX: BRIEF OVERVIEW OF THE TM

We present the basics of TM here. Those who already are familiar with the concept and notations of TM can ignore this appendix.

### A.1 BASIC CONCEPT OF THE TM

The input of a TM is denoted as $\mathbf{X} = [x_1, x_2, \ldots, x_o]$, where $x_k \in \{0, 1\}$, $k = 1, 2, \ldots, o$, and $o$ is the number of features. A literal is either the $x_k$ in the original form or its negation $\neg x_k$. A clause is a conjunction of literals, and each literal is associated with a TA. The TA is a 2-action learning automaton whose job is to decide whether to Include/Exclude its literal in/from the clause, and the decision is determined by the current state of the TA.

Figure 5 illustrates the structure of a TA with two actions and $2N$ states, where $N$ is the number of states for each action. This study considers $N$ as a finite number. When the TA is in any state between 0 to $N - 1$, the action "Include" is selected. The action becomes "Exclude" when the TA is in any state between $N$ to $2N - 1$. The transitions among the states are triggered by a reward or a penalty that the TA receives from the environment, which, in this case, is determined by different types of feedback defined in the TM (to be explained later).

A clause is associated with $2o$ TAs, forming a TA team. A TA team is denoted in general as $\mathcal{G}_j^i = \{\text{TA}_{k'}^{i,j} | 1 \leq k' \leq 2o\}$, where $k'$ is the index of the TA, $j$ is the index of the TA team/clause (multiple TA teams form a TM), and $i$ is the index of the TM/class to be identified (A TM identifies a class, multiple TMs identify multiple classes).

Suppose we are investigating the $i^{th}$ TM whose job is to identify class $i$, and that the TM is composed of $m$ TA teams. Then $C_j^i(\mathbf{X})$ can be used to denote the output of the $j^{th}$ TA team, which is a conjunctive clause:

$$\text{For training}: C_j^i(\mathbf{X}) = \begin{cases} \left( \bigwedge_{k \in \xi_j^i} x_k \right) \wedge \left( \bigwedge_{k \in \bar{\xi}_j^i} \neg x_k \right), & \text{for } \xi_j^i, \bar{\xi}_j^i \neq \emptyset, \\ 1, & \text{for } \xi_j^i, \bar{\xi}_j^i = \emptyset. \end{cases} \tag{12}$$

$$\text{For testing}: C_j^i(\mathbf{X}) = \begin{cases} \left( \bigwedge_{k \in \xi_j^i} x_k \right) \wedge \left( \bigwedge_{k \in \bar{\xi}_j^i} \neg x_k \right), & \text{for } \xi_j^i, \bar{\xi}_j^i \neq \emptyset, \\ 0, & \text{for } \xi_j^i, \bar{\xi}_j^i = \emptyset. \end{cases} \tag{13}$$

In Eqs. (12) and (13), $\xi_j^i$ and $\bar{\xi}_j^i$ are defined as the sets of indexes for the literals that have been included in the clause. $\xi_j^i$ contains the indexes of included original (non-negated) features, $x_k$, whereas $\bar{\xi}_j^i$ contains the indexes of included negated features, $\neg x_k$. Note that in propositional logic, an empty clause is typically defined as having a value of 1. However, empirical results indicate that TMs generally achieve higher test accuracy on new data when empty clauses are 0-valued. Therefore, during TM training, an "empty" clause outputs 1 to encourage the TAs to include literals, following the feedback mechanisms of the TM. In contrast, during TM testing, an "empty" clause outputs 0, indicating that it does not influence the final classification decision since it does not represent any specific sub-pattern.

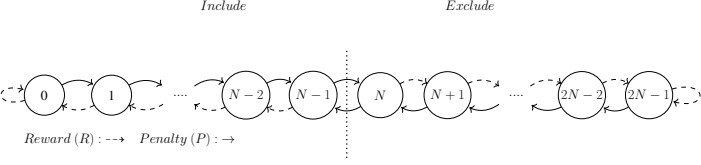

Figure 5: A two-action Tsetlin automaton with $2N$ states Jiao et al. (2022).

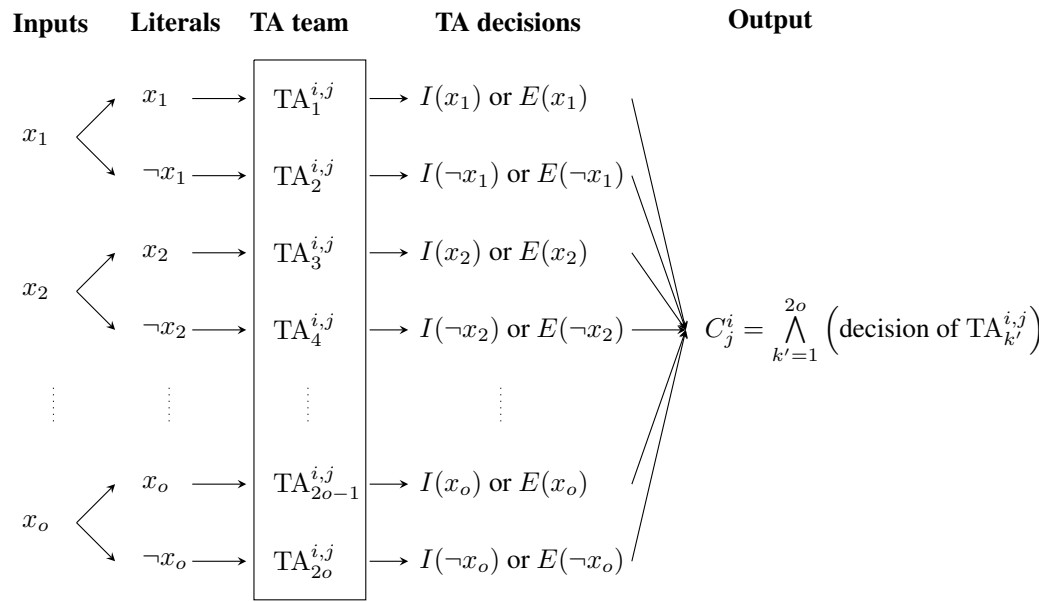

Figure 6: A TA team $G_j^i$ consisting of $2o$ TAs Zhang et al. (2022). Here $I(x_1)$ means "include $x_1$" and $E(x_1)$ means "exclude $x_1$".

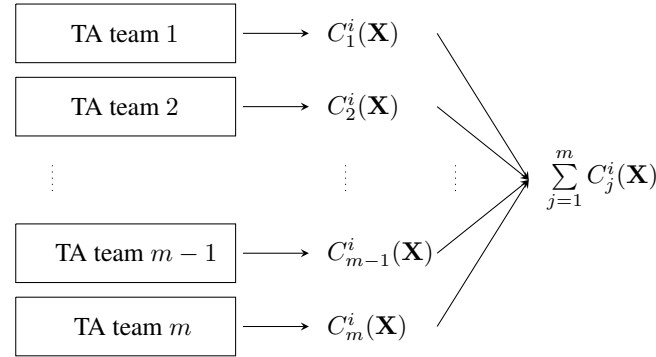

Figure 7: TM voting architecture Jiao et al. (2022).

Figure 6 illustrates the structure of a clause and its relationship to its literals. Here, for ease of notation, we define $I(x) = x$, $I(\neg x) = \neg x$, and $E(x) = E(\neg x) = 1$ in the analysis of the training procedure, with the latter meaning that an excluded literal does not contribute to the output.

Multiple clauses, i.e., the TA teams in conjunctive form, are assembled into a complete TM. There are two architectures for clause assembling: Disjunctive Normal Form Architecture and Voting Architecture. In this study, we focus on the latter one, as shown in Figure 7. The voting consists of summing the outputs of the clauses:

$$f_{\sum}(\mathcal{C}^i(\mathbf{X})) = \sum_{j=1}^{m} C_j^i(\mathbf{X}), \tag{14}$$

where $\mathcal{C}^i(\mathbf{X})$ is the set of trained clauses for class $i$.

The output of the TM, in turn, is decided by the unit step function:

$$\hat{y}^i = \begin{cases} 0 & \text{for } f_{\sum}(\mathcal{C}^i(\mathbf{X})) < Th \\ 1 & \text{for } f_{\sum}(\mathcal{C}^i(\mathbf{X})) \geq Th \end{cases}, \tag{15}$$

where $Th$ is a predefined threshold for classification. For example, the classifier $(x_1 \wedge \neg x_2) + (\neg x_1 \wedge x_2)$ captures the XOR-relation given $Th = 1$, meaning if any sub-pattern is satisfied, the input will be identified as following the XOR logic.

Note that for the voting architecture, the TM can assign a polarity to each TA team (Granmo, 2018). Specifically, TA teams with odd indices have positive polarity, learning from training samples with label 1, while those with even indices have negative polarity, learning from training samples with label 0. The only difference between these polarities is that the output of a clause associated with an even-indexed TA team will be flipped to its negative. The voting consists of summing the polarized clause outputs, and the threshold $Th$ is set to zero. For example, for the XOR operator with four clauses, the learned clauses with positive polarity can be $C_1 = x_1 \wedge \neg x_2$ and $C_3 = \neg x_1 \wedge x_2$, while the ones with negative polarity can be $C_2 = x_1 \wedge x_2$ and $C_4 = \neg x_1 \wedge \neg x_2$. In this case, when the testing sample $[x_1 = 1, x_2 = 0]$ arrives, the sum of the clause values is 1. On the contrary, when the testing sample $[x_1 = 0, x_2 = 0]$ arrives, the sum of the clause values is $-1$. In this way, with $Th = 0$, the system's decision range and tolerance is expected to be larger.

In this study, we consider only positive polarity clauses. The reason is two-folds: firstly, in the AND/OR case, once the TM has learned out the pattern that outputs 1, it also has learned the pattern that outputs 0, as they are complementary. Therefore, the learning/reasoning process of TM can be explained from the perspective of learning the pattern that outputs 1. Secondly, for the sake of easy analysis and better understanding.

## A.2 Training Process of the TM

The training process is built on letting all the TAs take part in a decentralized game. Training data $(\mathbf{X} = [x_1, x_2, ..., x_o], \ y^i)$ is obtained from a data set $\mathcal{S}$, distributed according to the probability distribution $P(\mathbf{X}, y^i)$. In the game, each TA is guided by Type I Feedback and Type II Feedback defined in Table 3 and Table 4, respectively. Type I Feedback is triggered when the training sample has a positive label, i.e., $y^i = 1$, meaning that the sample belongs to class $i$. When the training sample is labeled as not belonging to class $i$, i.e., $y^i = 0$, Type II Feedback is utilized for generating feedback. Examples demonstrating TA state transitions per feedback tables can be found in Section 3.1 in (Zhang et al., 2022). In brief, Type I feedback is to reinforce true positive and Type II feedback is to fight against false negative.

The parameter, $s$, controls the granularity of the clauses and a larger $s$ encourages more literals to be included in each clause. A more detailed analysis on parameter $s$ can be found in (Zhang et al., 2022).

| Value of the clause $C_j^i(\mathbf{X})$ | | 1 | | 0 | |
|---|---|---|---|---|---|
| Value of the Literal $x_k/\neg x_k$ | | 1 | 0 | 1 | 0 |
| **Include Literal** | $P(\text{Reward})$ | $\frac{s-1}{s}$ | NA | 0 | 0 |
| | $P(\text{Inaction})$ | $\frac{1}{s}$ | NA | $\frac{s-1}{s}$ | $\frac{s-1}{s}$ |
| | $P(\text{Penalty})$ | 0 | NA | $\frac{1}{s}$ | $\frac{1}{s}$ |
| **Exclude Literal** | $P(\text{Reward})$ | 0 | $\frac{1}{s}$ | $\frac{1}{s}$ | $\frac{1}{s}$ |
| | $P(\text{Inaction})$ | $\frac{1}{s}$ | $\frac{s-1}{s}$ | $\frac{s-1}{s}$ | $\frac{s-1}{s}$ |
| | $P(\text{Penalty})$ | $\frac{s-1}{s}$ | 0 | 0 | 0 |

Table 3: Type I Feedback — Feedback upon receiving a sample with label $y = 1$, for a single TA to decide whether to Include or Exclude a given literal $x_k/\neg x_k$ into $C_j^i$. NA means not applicable (Granmo, 2018).

To avoid the situation that a majority of the TA teams learn only one sub-pattern (or a subset of sub-patterns) while ignore other sub-patterns, forming an incomplete representation[3], the hyperparameter T is used to regulate the resource allocation. If the votes, i.e., the summation $f_\sum(\mathcal{C}^i(\mathbf{X}))$, for a certain sub-pattern $\mathbf{X}$ already reach a total of $T$ or more, neither rewards nor penalties are provided to the TAs when more training samples of this particular sub-pattern are given. In this way, we

---

[3]For example, for the XOR operator, we should avoid the situation that a majority of TA teams learn sub-pattern $x_1 = 0$ and $x_2 = 1$ and ignore sub-pattern $x_1 = 1$ and $x_2 = 0$, making the learning outcome biased/unbalanced. A proper configuration of $T$ can avoid this situation.

| Value of the clause $C_j^i(\mathbf{X})$ | | 1 | | 0 | |
|---|---|---|---|---|---|
| Value of the Literal $x_k/\neg x_k$ | | 1 | 0 | 1 | 0 |
| **Include Literal** | $P(\text{Reward})$ | 0 | NA | 0 | 0 |
| | $P(\text{Inaction})$ | 1.0 | NA | 1.0 | 1.0 |
| | $P(\text{Penalty})$ | 0 | NA | 0 | 0 |
| **Exclude Literal** | $P(\text{Reward})$ | 0 | 0 | 0 | 0 |
| | $P(\text{Inaction})$ | 1.0 | 0 | 1.0 | 1.0 |
| | $P(\text{Penalty})$ | 0 | 1.0 | 0 | 0 |

Table 4: Type II Feedback — Feedback upon receiving a sample with label $y = 0$, for a single TA to decide whether to Include or Exclude a given literal $x_k/\neg x_k$ into $C_j^i$. NA means not applicable (Granmo, 2018).

can ensure that each specific sub-pattern can be captured by a limited number, i.e., $T$, of available clauses, allowing sparse sub-pattern representations among competing sub-patterns. In more details, the strategy works as follows:

**Generating Type I Feedback.** If the label of the training sample $\mathbf{X}$ is $y^i = 1$, we generate, in probability, *Type I Feedback* for each clause $C_j^i \in \mathcal{C}^i$. The probability of generating Type I Feedback is (Granmo, 2018):

$$u_1 = \frac{T - \max(-T, \min(T, f_{\sum}(\mathcal{C}^i(\mathbf{X}))))}{2T}. \tag{16}$$

**Generating Type II Feedback.** If the lable of the training sample $\mathbf{X}$ is $y^i = 0$, we generate, again, in probability, *Type II Feedback* to each clause $C_j^i \in \mathcal{C}^i$. The probability is (Granmo, 2018):

$$u_2 = \frac{T + \max(-T, \min(T, f_{\sum}(\mathcal{C}^i(\mathbf{X}))))}{2T}. \tag{17}$$

After Type I Feedback or Type II Feedback is generated for a clause, each individual TA within each clause is given reward/penalty/inaction according to the probability defined, and then the states of the corresponding TAs are updated.

# B  APPENDIX: DETAILED PROOF OF THE CONVERGENCE OF THE AND OPERATOR

**Proof:** In this Appendix, we will prove Theorem 1. The condition $u_1 > 0$ and $u_2 > 0$ guarantees that all types of samples for AND operator, following Eq. (18), are always given and no specific type is blocked during training. The goal of the proof is to show that the system transitions will guarantee the actions of $TA_1$, $TA_2$, $TA_3$, and $TA_4$ to be I, E, I, E, and these actions correspond to the unique absorbing state of the system.

$$P(y = 1|x_1 = 1, x_2 = 1) = 1, \tag{18}$$
$$P(y = 0|x_1 = 0, x_2 = 1) = 1,$$
$$P(y = 0|x_1 = 1, x_2 = 0) = 1,$$
$$P(y = 0|x_1 = 0, x_2 = 0) = 1.$$

In Subsections B.1, we will describe the transitions of the system in an exhaustive manner. Thereafter, in the Subsection B.2, we summarize the transitions in Subsection B.1 and reveal the absorbing state of the system, which is the intended AND operator.

## B.1  THE TRANSITIONS OF THE TAS

In order to analyze the transitions of the system, we freeze the transition of the two TAs for the first bit of the input and study the transition of the second bit of input. Clearly, there are four cases for the first bit, $x_1$, as:

- Case 1: $TA_1$ = E, $TA_2$ = I, i.e., include $\neg x_1$.
- Case 2: $TA_1$ = I, $TA_2$ = E, i.e., include $x_1$.
- Case 3: $TA_1$ = E, $TA_2$ = E, i.e., exclude both $x_1$ and $\neg x_1$.
- Case 4: $TA_1$ = I, $TA_2$ = I, i.e., include both $x_1$ and $\neg x_1$.

In what follows, we will analyze the transition of the TAs for $x_2$, given the TAs of $x_1$ frozen in the above four distinct cases, one by one.

### B.1.1  CASE 1: INCLUDE $\neg x_1$

In this subsection, we assume that the TAs for first bit is frozen as $TA_1$ = E and $TA_2$ = I, and thus the overall joint actions of TAs for the first bit give "$\neg x_1$". In this case, we have 4 situations to study, detailed below:

- Situation1: We study the transition of $TA_3$ when it has "Include" as its current action, given different actions of $TA_4$ (i.e., when the action of $TA_4$ is frozen as "Include" or "Exclude".).

- Situation 2: We study the transition of $TA_3$ when it has "Exclude" as its current action, given different actions of $TA_4$ (i.e., when the action of $TA_4$ is frozen as "Include" or "Exclude".).

- Situation 3: We study the transition of $TA_4$ when it has "Include" as its current action, given different actions of $TA_3$ (i.e., when the action of $TA_3$ is frozen as "Include" or "Exclude".).

- Situation 4: We study the transition of $TA_4$ when it has "Exclude" as its current action, given different actions of $TA_3$ (i.e., when the action of $TA_3$ is frozen as "Include" or "Exclude".).

In what follows, we will go through, exhaustively, the four situations.

*B.1.1.1  Study $TA_3$ with Action Include*

Here we study the transitions of $TA_3$ when its current action is *Include*, given different actions of $TA_4$ and input samples. For ease of expressions, the self-loops of the transitions are not depicted

in the transition diagram. Clearly, this situation has 8 instances, depending on the variations of the training samples and the status of $TA_4$, where the first four correspond to the instances with $TA_4 = E$ while the remaining four represent the instances with $TA_4 = I$.

Now we study the first instance, with $x_1 = 1$, $x_2 = 1$, $y = 1$, and $TA_4 = E$. Clearly, this training sample will trigger Type I feedback because $y = 1$. Together with the current status of the other TAs, the clause is determined to be $C = \neg x_1 \wedge x_2 = 0$ and the literal is $x_2 = 1$. From Table 3, we know that the penalty probability is $\frac{1}{s}$ and the inaction probability is $\frac{s-1}{s}$. To indicate the transitions, we have plotted the diagram, with the transitions for penalty ($P$) below. Note that the overall transition probability is $u_1\frac{1}{s}$, where $u_1$ is defined in Eq. (3). Here, we have assumed $u_1 > 0$.

Condition: $x_1 = 1$, $x_2 = 1$, $y = 1$,
$TA_4 = E$.
Thus, Type I, $x_2 = 1$,
$C = \neg x_1 \wedge x_2 = 0$.

We here continue with analyzing another example shown below. In this instance, it covers the training samples: $x_1 = 1$, $x_2 = 0$, $y = 0$, and $TA_4 = E$. Clearly, the training sample will trigger Type II feedback because $y = 0$. The clause output becomes $C_3 = \neg x_1 \wedge x_2 = 0$. Because we now study $TA_3$, the corresponding literal is $x_2 = 0$. Based on the information above, we can check from Table 4 and find the probability of "Inaction" is 1. For this reason, the transition diagram does not have any arrow, indicating that there is "No transition" for $TA_3$.

Condition: $x_1 = 1$, $x_2 = 0$, $y = 0$,
$TA_4 = E$.
Thus, Type II, $x_2 = 0$,
$C = \neg x_1 \wedge x_2 = 0$.

No transition

The same analytical principle applies for all the other instances, and we therefore will not explain them in detail. Instead, we just list the transition diagrams.

Condition: $x_1 = 0$, $x_2 = 1$, $y = 0$,
$TA_4 = E$.
Thus, Type II, $x_2 = 1$,
$C = \neg x_1 \wedge x_2 = 1$.

No transition

Condition: $x_1 = 0$, $x_2 = 0$, $y = 0$,
$TA_4 = E$.
Thus, Type II, $x_2 = 0$,
$C = \neg x_1 \wedge x_2 = 0$.

No transition

Condition: $x_1 = 1$, $x_2 = 1$, $y = 1$,
$TA_4 = I$.
Thus, Type I, $x_2 = 1$,
$C = \neg x_1 \wedge x_2 \wedge \neg x_2 = 0$.

Condition: $x_1 = 1$, $x_2 = 0$, $y = 0$,
$TA_4 = I$.
Thus, Type II, $x_2 = 0$,
$C = \neg x_1 \wedge x_2 \wedge \neg x_2 = 0$.

No transition

Condition: $x_1 = 0$, $x_2 = 1$, $y = 0$,
$TA_4 = I$.
Thus, Type II, $x_2 = 1$,
$C = \neg x_1 \wedge x_2 \wedge \neg x_2 = 0$.

No transition

Condition: $x_1 = 0$, $x_2 = 0$, $y = 0$,
TA$_4$ = I.
Thus, Type II, $x_2 = 0$,
$C = \neg x_1 \wedge x_2 \wedge \neg x_2 = 0$.

No transition

### B.1.1.2  Study TA$_3$ with Action Exclude

Here we study the transitions of TA$_3$ when its current action is *Exclude*, given different actions of TA$_4$ and input samples. This situation has 8 instances, depending on the variations of the training samples and the status of TA$_4$. In this subsection and the following subsections, we will not plot the transition diagrams for "No transition".

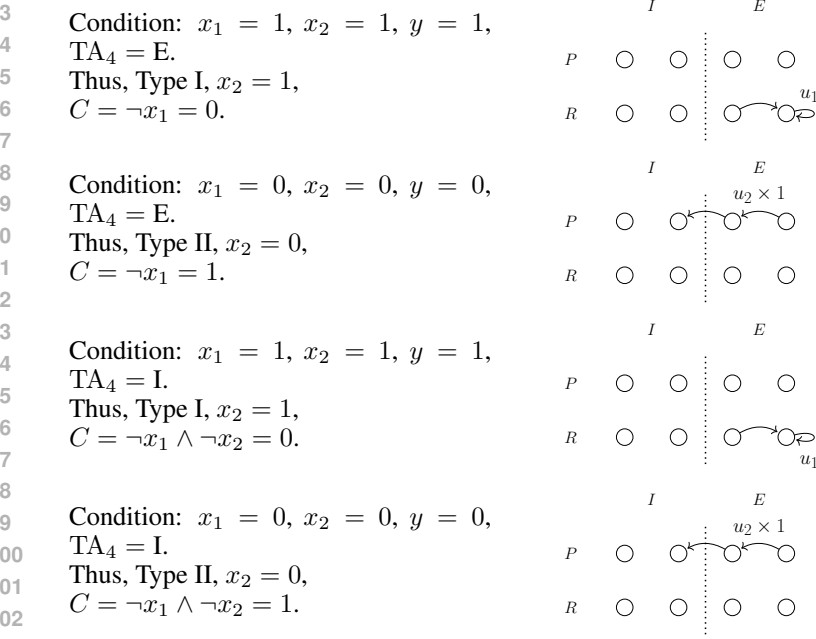

Condition: $x_1 = 1$, $x_2 = 1$, $y = 1$,
TA$_4$ = E.
Thus, Type I, $x_2 = 1$,
$C = \neg x_1 = 0$.

Condition: $x_1 = 0$, $x_2 = 0$, $y = 0$,
TA$_4$ = E.
Thus, Type II, $x_2 = 0$,
$C = \neg x_1 = 1$.

Condition: $x_1 = 1$, $x_2 = 1$, $y = 1$,
TA$_4$ = I.
Thus, Type I, $x_2 = 1$,
$C = \neg x_1 \wedge \neg x_2 = 0$.

Condition: $x_1 = 0$, $x_2 = 0$, $y = 0$,
TA$_4$ = I.
Thus, Type II, $x_2 = 0$,
$C = \neg x_1 \wedge \neg x_2 = 1$.

### B.1.1.3  Study TA$_4$ with Action Include

Here we list the transitions for TA$_4$ when its current action is *Include.*

Condition: $x_1 = 1$, $x_2 = 1$, $y = 1$,
TA$_3$ = E.
Thus, Type I, $\neg x_2 = 0$,
$C = \neg x_1 \wedge \neg x_2 = 0$.

Condition: $x_1 = 1$, $x_2 = 1$, $y = 1$,
TA$_3$ = I.
Thus, Type I, $\neg x_2 = 0$,
$C = \neg x_1 \wedge x_2 \wedge \neg x_2 = 0$.

### B.1.1.4  Study TA$_4$ with Action Exclude

Here we list the transitions for TA$_4$ when its current action is *Exclude.*

Condition: $x_1 = 1$, $x_2 = 1$, $y = 1$,
TA$_3$ = E.
Thus, Type I, $\neg x_2 = 0$,
$C = \neg x_1 = 0$.

Condition: $x_1 = 0$, $x_2 = 1$, $y = 0$,
$\text{TA}_3 = \text{E}$.
Thus, Type II, $\neg x_2 = 0$,
$C = \neg x_1 = 1$.

Condition: $x_1 = 1$, $x_2 = 1$, $y = 1$,
$\text{TA}_3 = \text{I}$.
Thus, Type I, $\neg x_2 = 0$,
$C = \neg x_1 \wedge x_2 = 0$.

Condition: $x_1 = 0$, $x_2 = 1$, $y = 0$,
$\text{TA}_3 = \text{I}$.
Thus, Type II, $\neg x_2 = 0$,
$C = \neg x_1 \wedge x_2 = 1$.

### B.1.2 CASE 2: INCLUDE $x_1$

For Case 2, we assume that the actions of the TAs for the first bit are frozen as $\text{TA}_1 = \text{I}$ and $\text{TA}_2 = \text{E}$, and thus the overall joint action for the first bit is "$x_1$". Similar to Case 1, we also have 4 situations.

*B.1.2.1 Study $\text{TA}_3$ with Action Include*

Condition: $x_1 = 1$, $x_2 = 1$, $y = 1$,
$\text{TA}_4 = \text{E}$.
Thus, Type I, $x_2 = 1$,
$C = x_1 \wedge x_2 = 1$.

Condition: $x_1 = 1$, $x_2 = 1$, $y = 1$,
$\text{TA}_4 = \text{I}$.
Thus, Type I, $x_2 = 1$,
$C = x_1 \wedge x_2 \wedge \neg x_2 = 0$.

*B.1.2.2 Study $\text{TA}_3$ with Action Exclude*

Condition: $x_1 = 1$, $x_2 = 1$, $y = 1$,
$\text{TA}_4 = \text{E}$.
Thus, Type I, $x_2 = 1$,
$C = x_1 = 1$.

Condition: $x_1 = 1$, $x_2 = 0$, $y = 0$,
$\text{TA}_4 = \text{E}$.
Thus, Type II, $x_2 = 0$,
$C = x_1 = 1$.

Condition: $x_1 = 1$, $x_2 = 1$, $y = 1$,
$\text{TA}_4 = \text{I}$.
Thus, Type I, $x_2 = 1$,
$C = x_1 \wedge \neg x_2 = 0$.

Condition: $x_1 = 1$, $x_2 = 0$, $y = 0$,
$\text{TA}_4 = \text{I}$.
Thus, Type II, $x_2 = 0$,
$C = x_1 \wedge \neg x_2 = 1$.

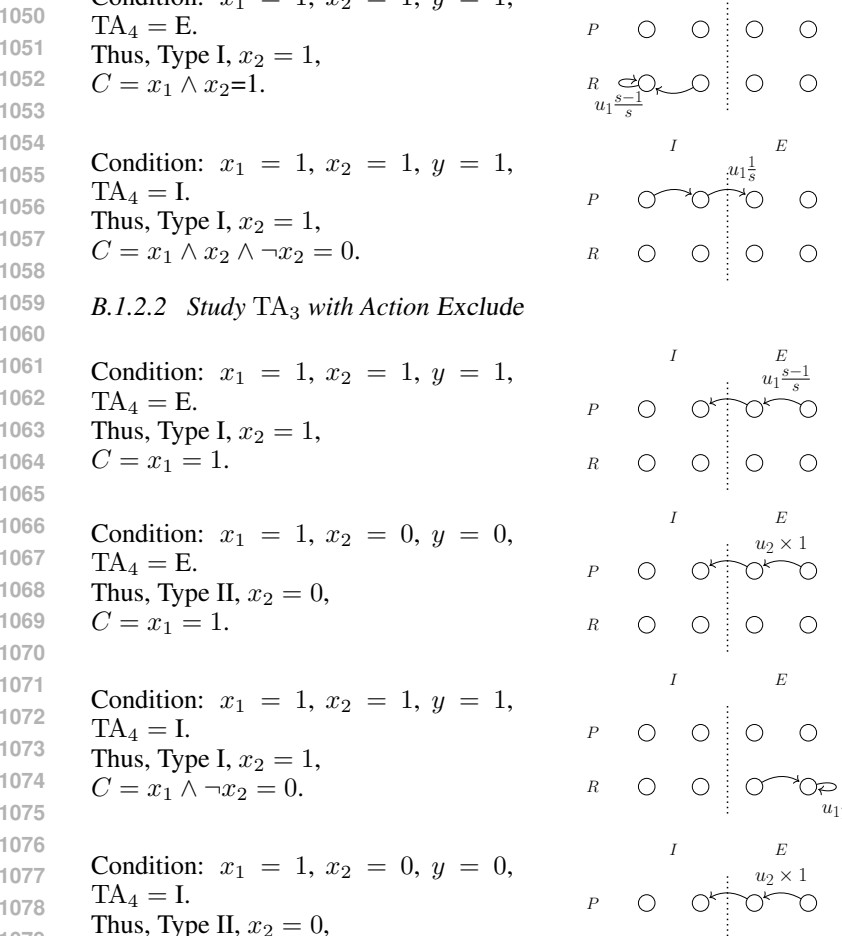

*B.1.2.3  Study* $TA_4$ *with Action Include*

Condition: $x_1 = 1$, $x_2 = 1$, $y = 1$,
$TA_3 = E$.
Thus, Type I, $\neg x_2 = 0$,
$C = x_1 \wedge \neg x_2 = 0$.

Condition: $x_1 = 1$, $x_2 = 1$, $y = 1$,
$TA_3 = I$.
Thus, Type I, $\neg x_2 = 0$,
$C = x_1 \wedge x_2 \wedge \neg x_2 = 0$.

*B.1.2.4  Study* $TA_4$ *with Action Exclude*

Condition: $x_1 = 1$, $x_2 = 1$, $y = 1$,
$TA_3 = E$.
Thus, Type I, $\neg x_2 = 0$,
$C = x_1 = 1$.

Condition: $x_1 = 1$, $x_2 = 1$, $y = 1$,
$TA_3 = I$.
Thus, Type I, $\neg x_2 = 0$,
$C = x_1 \wedge x_2 = 1$.

## B.1.3   CASE 3: EXCLUDE BOTH $\neg x_1$ AND $x_1$

For Case 3, we assume that the actions of TAs for the first bit are frozen as $TA_1 = E$ and $TA_2 = E$, with 4 situations. Note that in the training process, when all literals are excluded, $C$ is assigned to 1.

*B.1.3.1  Study* $TA_3$ *with Action Include*

Condition: $x_1 = 1$, $x_2 = 1$, $y = 1$,
$TA_4 = E$.
Thus, Type I, $x_2 = 1$,
$C = x_2 = 1$.

Condition: $x_1 = 1$, $x_2 = 1$, $y = 1$,
$TA_4 = I$.
Thus, Type I, $x_2 = 1$,
$C = 0$.

*B.1.3.2  Study* $TA_3$ *with Action Exclude*

Condition: $x_1 = 1$, $x_2 = 1$, $y = 1$,
$TA_4 = E$.
Thus, Type I, $x_2 = 1$,
$C = 1$.

Condition: $x_1 = 1$, $x_2 = 0$, $y = 0$,
$TA_4 = E$.
Thus, Type II, $x_2 = 0$,
$C = 1$.

Condition: $x_1 = 0$, $x_2 = 0$, $y = 0$, $\text{TA}_4 = \text{E}$.
Thus, Type II, $x_2 = 0$,
$C = 1$.

$I \qquad E$
$u_1 \times 1$
$P$
$R$

Condition: $x_1 = 1$, $x_2 = 1$, $y = 1$, $\text{TA}_4 = \text{I}$.
Thus, Type I, $x_2 = 1$,
$C = 0$.

$I \qquad E$
$P$
$R$
$u_{1\frac{1}{s}}$

Condition: $x_1 = 1$, $x_2 = 0$, $y = 0$, $\text{TA}_4 = \text{I}$.
Thus, Type II, $x_2 = 0$,
$C = 1$.

$I \qquad E$
$u_2 \times 1$
$P$
$R$

Condition: $x_1 = 0$, $x_2 = 0$, $y = 0$, $\text{TA}_4 = \text{I}$.
Thus, Type II, $x_2 = 0$,
$C = 1$.

$I \qquad E$
$u_2 \times 1$
$P$
$R$

### B.1.3.3  Study $\text{TA}_4$ with Action Include

Condition: $x_1 = 1$, $x_2 = 1$, $y = 1$, $\text{TA}_3 = \text{E}$.
Thus, Type I, $\neg x_2 = 0$,
$C = \neg x_2 = 0$.

$I \qquad E$
$u_{1\frac{1}{s}}$
$P$
$R$

Condition: $x_1 = 1$, $x_2 = 1$, $y = 1$, $\text{TA}_3 = \text{I}$.
Thus, Type I, $\neg x_2 = 0$,
$C = \neg x_2 \wedge x_2 = 0$.

$I \qquad E$
$u_{1\frac{1}{s}}$
$P$
$R$

### B.1.3.4  Study $\text{TA}_4$ with Action Exclude

Condition: $x_1 = 1$, $x_2 = 1$, $y = 1$, $\text{TA}_3 = \text{E}$.
Thus, Type I, $\neg x_2 = 0$,
$C = 1$.

$I \qquad E$
$P$
$R$
$u_{1\frac{1}{s}}$

Condition: $x_1 = 0$, $x_2 = 1$, $y = 0$, $\text{TA}_3 = \text{E}$.
Thus, Type II, $\neg x_2 = 0$,
$C = 1$.

$I \qquad E$
$u_2 \times 1$
$P$
$R$

Condition: $x_1 = 1$, $x_2 = 1$, $y = 1$, $\text{TA}_3 = \text{I}$.
Thus, Type I, $\neg x_2 = 0$,
$C = 1$.

$I \qquad E$
$P$
$R$
$u_{1\frac{1}{s}}$

Condition: $x_1 = 0$, $x_2 = 1$, $y = 0$, $\text{TA}_3 = \text{I}$.
Thus, Type II, $\neg x_2 = 0$,
$C = 1$.

$I \qquad E$
$u_2 \times 1$
$P$
$R$

### B.1.4 CASE 4: INCLUDE BOTH $\neg x_1$ AND $x_1$

For Case 4, we assume that the actions of TAs for the first bit are frozen as $TA_1 = I$ and $TA_2 = I$, and thus $C = \mathbf{0}$ **always**. Similarly, we also have 4 situations, detailed below.

*B.1.4.1 Study $TA_3$ with Action Include*

Condition: $x_1 = 1$, $x_2 = 1$, $y = 1$,
$TA_4 = E$.
Thus, Type I, $x_2 = 1$,
$C = 0$.

Condition: $x_1 = 1$, $x_2 = 1$, $y = 1$,
$TA_4 = I$.
Thus, Type I, $x_2 = 1$,
$C = 0$.

*B.1.4.2 Study $TA_3$ with Action Exclude*

Condition: $x_1 = 1$, $x_2 = 1$, $y = 1$,
$TA_4 = E$.
Thus, Type I, $x_2 = 1$,
$C = 0$.

Condition: $x_1 = 1$, $x_2 = 1$, $y = 1$,
$TA_4 = I$.
Thus, Type I, $x_2 = 1$,
$C = 0$.

*B.1.4.3 Study $TA_4$ with Action Include*

Condition: $x_1 = 1$, $x_2 = 1$, $y = 1$,
$TA_3 = E$.
Thus, Type I, $\neg x_2 = 0$,
$C = 0$.

Condition: $x_1 = 1$, $x_2 = 1$, $y = 1$,
$TA_3 = I$.
Thus, Type I, $\neg x_2 = 0$,
$C = 0$.

*B.1.4.4 Study $TA_4$ with Action Exclude*

Condition: $x_1 = 1$, $x_2 = 1$, $y = 1$,
$TA_3 = E$.
Thus, Type I, $\neg x_2 = 0$,
$C = 0$.

Condition: $x_1 = 1$, $x_2 = 1$, $y = 1$,
$TA_3 = I$.
Thus, Type I, $\neg x_2 = 0$,
$C = 0$.

So far, we have gone through, exhaustively, the transitions of $TA_3$ and $TA_4$ for all the cases (all possible training samples and system states). Hereafter, we can summarize the direction of transitions

and study the convergence properties of the system for the given training samples, to be detailed in the next subsection.

### B.2 SUMMARIZE OF THE DIRECTIONS OF TRANSITIONS IN DIFFERENT CASES

Based on the analysis above, we summarize here what happens to $TA_3$ and $TA_4$, given different status (Cases) of $TA_1$ and $TA_2$. More specifically, we will summarize here the directions of the transitions for the TAs. For example, "$TA_3 \Rightarrow E$" means that $TA_3$ will move towards the action "Exclude", while "$TA_4 \Rightarrow E$ or I" means $TA_4$ transits towards either "Exclude" or "Include".

**Scenario 1:** Study $TA_3 = I$ and $TA_4 = I$.

| **Case 1**, we have: | **Case 3**, we have: |
|---|---|
| $TA_3 \Rightarrow E$. | $TA_3 \Rightarrow E$. |
| $TA_4 \Rightarrow E$. | $TA_4 \Rightarrow E$. |
| **Case 2**, we have: | **Case 4**, we have: |
| $TA_3 \Rightarrow E$. | $TA_3 \Rightarrow E$. |
| $TA_4 \Rightarrow E$. | $TA_4 \Rightarrow E$. |

From the facts presented above, we can confirm that regardless the state of $TA_1$ and $TA_2$, if $TA_3 = I$ and $TA_4 = I$, they ($TA_3$ and $TA_4$) will eventually move out of their states.

**Scenario 2:** Study $TA_3 = I$ and $TA_4 = E$.

| **Case 1**, we have: | **Case 3**, we have: |
|---|---|
| $TA_3 \Rightarrow E$. | $TA_3 \Rightarrow I$. |
| $TA_4 \Rightarrow E$ or I. | $TA_4 \Rightarrow E$ or I. |
| **Case 2**, we have: | **Case 4**, we have: |
| $TA_3 \Rightarrow I$. | $TA_3 \Rightarrow E$. |
| $TA_4 \Rightarrow E$. | $TA_4 \Rightarrow E$. |

For Scenario 2 Case 2, we can observe that if $TA_3 = I$, $TA_4 = E$, $TA_1 = I$, and $TA_2 = E$, $TA_3$ will move deeper to "include" and $TA_4$ will go deeper to "exclude". It is not difficult to derive also that $TA_1$ will move deeper to "include" and $TA_2$ will transfer deeper to "exclude" in this circumstance. This tells us that the TAs in states $TA_3 = I$, $TA_4 = E$, $TA_1 = I$, and $TA_2 = E$, reinforce each other to move deeper to their corresponding directions and they therefore construct an absorbing state of the system. If it is the only absorbing state, we can conclude that the TM converge to the intended "AND" operation.

In Scenario 2, we can observe for Cases 1, 3, and 4, the actions for $TA_3$ and $TA_4$ are not absorbing because the TAs will not be reinforced to move monotonically deeper to the states of the corresponding actions for difference cases.

For Scenario 2, Case 3, $TA_4$ has two possible directions to transit, I or E, depending on the input variables of the training sample. For action exclude, it will be reinforced when training sample $x_1 = 1$ and $x_2 = 1$ is given, based on Type I feedback. However, $TA_4$ will transit towards "include" side when training sample $x_1 = 0$ and $x_2 = 1$ is given, due to Type II feedback. Therefore, the direction of the transition for $TA_4$ is I or E, depending on the training samples. In the following paragraphs, when "or" appears in the transition direction, the same concept applies.

**Scenario 3:** Study $TA_3 = E$ and $TA_4 = I$.

| **Case 1**, we have: | **Case 3**, we have: |
|---|---|
| $TA_3 \Rightarrow E$ or I. | $TA_3 \Rightarrow E$ or I. |
| $TA_4 \Rightarrow E$. | $TA_4 \Rightarrow E$. |
| **Case 2**, we have: | **Case 4**, we have: |
| $TA_3 \Rightarrow E$ or I. | $TA_3 \Rightarrow E$. |
| $TA_4 \Rightarrow E$. | $TA_4 \Rightarrow E$. |

In Scenario 3, we can see that the actions for $TA_3 = E$ and $TA_4 = I$ are not absorbing because the TAs will not be reinforced to move deeper to the states of the corresponding actions.

**Scenario 4:** Study $TA_3 = E$ and $TA_4 = E$.

**Case 1**, we have:    **Case 3**, we have:
$TA_3 \Rightarrow I$ or E.    $TA_3 \Rightarrow I$.
$TA_4 \Rightarrow I$ or E.    $TA_4 \Rightarrow I$ or E.
**Case 2**, we have:    **Case 4**, we have:
$TA_3 \Rightarrow I$.    $TA_3 \Rightarrow E$.
$TA_4 \Rightarrow E$.    $TA_4 \Rightarrow E$.

In Scenario 4, we see that, the actions for $TA_3 = E$ and $TA_4 = E$ seem to be an absorbing state, because the states of TAs will move deeper in Case 4. After a revisit of the condition for Case 4, i.e., include both $\neg x_1$ and $x_1$, we understand that this condition is not absorbing. In fact, when $TA_1$ and $TA_2$ both have "Include" as their actions, they monotonically move towards "Exclude". Therefore, from the overall system's perspective, the system state $TA_1 = I$, $TA_2 = I$, $TA_3 = E$, and $TA_4 = E$ is not absorbing. For the other cases in this scenario, there is no absorbing state.

Based on the above analysis, we understand that there is only one absorbing condition in the system, namely, $TA_1 = I$, $TA_2 = E$, $TA_3 = I$, and $TA_4 = E$, for the given training samples with AND logic. The same conclusion applies when we freeze the transition of the two TAs for the second bit of the input and study behavior of the first bit of input. Therefore, we can conclude that the TM with a clause can learn to be the intended AND operator, almost surely, in infinite time horizon. We thus complete the proof of Theorem 1. ∎

## C    PROOF OF LEMMA 1

The probability of the training samples for the noise-free OR operator can be presented by the following equations.

$$P(y = 1|x_1 = 1, x_2 = 1) = 1, \tag{19}$$
$$P(y = 1|x_1 = 0, x_2 = 1) = 1,$$
$$P(y = 1|x_1 = 1, x_2 = 0) = 1,$$
$$P(y = 0|x_1 = 0, x_2 = 0) = 1.$$

Clearly, there are three sub-patterns of $x_1$ and $x_2$ that will give $y = 1$, i.e., $[x_1 = 1, \ x_2 = 1]$, $[x_1 = 1, \ x_2 = 0]$, and $[x_1 = 0, \ x_2 = 1]$. More specifically, Eq. (19) can be split into three cases, corresponding to the three sub-patterns:

$$P(y = 1|x_1 = 1, x_2 = 1) = 1, \tag{20}$$
$$P(y = 0|x_1 = 0, x_2 = 0) = 1,$$

$$P(y = 1|x_1 = 0, x_2 = 1) = 1, \tag{21}$$
$$P(y = 0|x_1 = 0, x_2 = 0) = 1,$$

and

$$P(y = 1|x_1 = 1, x_2 = 0) = 1, \tag{22}$$
$$P(y = 0|x_1 = 0, x_2 = 0) = 1.$$

In what follows, we will show the convergence of the three sub-patterns, i.e., Lemma 1.

The convergence analyses of the above three sub-patterns can be derived by reusing the analyses of the sub-patterns of the XOR operator plus the AND operator. For the sub-pattern described by Eq. (20), we can confirm that the TAs will indeed converge to $TA_1 = I$, $TA_2 = E$, $TA_3 = I$, and $TA_4 = E$, by studying the transition diagrams in Subsection B when input samples of $[x_1 = 0, x_2 = 1]$ and $[x_1 = 1, x_2 = 0]$ are removed. In this case, the directions of the transitions for different scenarios are summarized below.

**Scenario 1:** Study $TA_3 = I$ and $TA_4 = I$.

**Case 1**, we have:       **Case 3**, we have:
$TA_3 \Rightarrow E$.       $TA_3 \Rightarrow E$.
$TA_4 \Rightarrow E$.       $TA_4 \Rightarrow E$.
**Case 2**, we have:       **Case 4**, we have:
$TA_3 \Rightarrow E$.       $TA_3 \Rightarrow E$.
$TA_4 \Rightarrow E$.       $TA_4 \Rightarrow E$.

**Scenario 2:** Study $TA_3 = I$ and $TA_4 = E$.

**Case 1**, we have:       **Case 3**, we have:
$TA_3 \Rightarrow E$.       $TA_3 \Rightarrow I$.
$TA_4 \Rightarrow E$.       $TA_4 \Rightarrow E$.
**Case 2**, we have:       **Case 4**, we have:
$TA_3 \Rightarrow I$.       $TA_3 \Rightarrow E$.
$TA_4 \Rightarrow E$.       $TA_4 \Rightarrow E$.

**Scenario 3:** Study $TA_3 = E$ and $TA_4 = I$.

**Case 1**, we have:       **Case 3**, we have:
$TA_3 \Rightarrow E$ or $I$.       $TA_3 \Rightarrow E$ or $I$.
$TA_4 \Rightarrow E$.       $TA_4 \Rightarrow E$.
**Case 2**, we have:       **Case 4**, we have:
$TA_3 \Rightarrow E$.       $TA_3 \Rightarrow E$.
$TA_4 \Rightarrow E$.       $TA_4 \Rightarrow E$.

**Scenario 4:** Study $TA_3 = E$ and $TA_4 = E$.

| **Case 1**, we have: | **Case 3**, we have: |
|---|---|
| $TA_3 \Rightarrow$ I or E. | $TA_3 \Rightarrow$ I. |
| $TA_4 \Rightarrow$ E. | $TA_4 \Rightarrow$ E. |
| **Case 2**, we have: | **Case 4**, we have: |
| $TA_3 \Rightarrow$ I. | $TA_3 \Rightarrow$ E. |
| $TA_4 \Rightarrow$ E. | $TA_4 \Rightarrow$ E. |

Comparing the analysis with the one in Subsection B.2, there is apparently another possible absorbing case, which can be observed in Scenario 2, Case 3, where $TA_3 = I$ and $TA_4 = E$, given $TA_1 = E$ and $TA_2 = E$. However, given $TA_3 = I$ and $TA_4 = E$, the TAs for the first bit, i.e., $TA_1 = E$ and $TA_2 = E$, will not move only towards Exclude. Therefore, they do not reinforce each other to move to deeper states for their current actions. For this reason, the system in $TA_3 = I$, $TA_4 = E$, $TA_1 = E$, and $TA_2 = E$, is not in an absorbing state. In addition, given $TA_3 = I$ and $TA_4 = E$, $TA_1$ and $TA_2$ with actions E and E will transit towards I and E, encouraging the overall system to move towards I, E, I, and E. Consequently, the system state with $TA_1 = I$, $TA_2 = E$, $TA_3 = I$, and $TA_4 = E$ is still the only absorbing case for the given training samples following Eq. (20).

For Eq. (21), similar to the proof of in Lemma 1 in (Jiao et al., 2022), we can derive that the TAs will converge in $TA_1 = E$, $TA_2 = I$, $TA_3 = I$, and $TA_4 = E$. The transition diagrams for the samples of Eq. (21) are in fact a subset of the ones presented in Subsection 3.2.1 and Appendix 2 of (Jiao et al., 2022), when the input samples of $[x_1 = 1$ and $x_2 = 1]$ are removed. We summarize below only the directions of transitions.

The directions of the transitions of the TAs for the second input bit, i.e., $x_2/\neg x_2$, when the TAs for the first input bit are frozen, are summarized as follows (based on the subset of the transition diagrams in Subsection 3.2.1 of (Jiao et al., 2022)).

**Scenario 1:** Study $TA_3 = I$ and $TA_4 = I$.

| **Case 1:** we have | **Case 3:** we have |
|---|---|
| $TA_3 \rightarrow$ E | $TA_3 \rightarrow$ E |
| $TA_4 \rightarrow$ E | $TA_4 \rightarrow$ E |
| **Case 2:** we have | **Case 4:** we have |
| $TA_3 \rightarrow$ E | $TA_3 \rightarrow$ E |
| $TA_4 \rightarrow$ E | $TA_4 \rightarrow$ E |

**Scenario 2:** Study $TA_3 = I$ and $TA_4 = E$.

| **Case 1:** we have | **Case 3:** we have |
|---|---|
| $TA_3 \rightarrow$ I | $TA_3 \rightarrow$ I |
| $TA_4 \rightarrow$ E | $TA_4 \rightarrow$ E |
| **Case 2:** we have | **Case 4:** we have |
| $TA_3 \rightarrow$ E | $TA_3 \rightarrow$ E |
| $TA_4 \rightarrow$ E | $TA_4 \rightarrow$ E |

**Scenario 3:** Study $TA_3 = E$ and $TA_4 = I$.

| **Case 1:** we have | **Case 3:** we have |
|---|---|
| $TA_3 \rightarrow$ I, or E | $TA_3 \rightarrow$ I, or E |
| $TA_4 \rightarrow$ E | $TA_4 \rightarrow$ E |
| **Case 2:** we have | **Case 4:** we have |
| $TA_3 \rightarrow$ E | $TA_3 \rightarrow$ E |
| $TA_4 \rightarrow$ E | $TA_4 \rightarrow$ E |

**Scenario 4:** Study $TA_3 = E$ and $TA_4 = E$.

**Case 1:** we have
$TA_3 \rightarrow I$
$TA_4 \rightarrow E$
**Case 2:** we have
$TA_3 \rightarrow E$
$TA_4 \rightarrow E$

**Case 3:** we have
$TA_3 \rightarrow I$
$TA_4 \rightarrow E$
**Case 4:** we have
$TA_3 \rightarrow E$
$TA_4 \rightarrow E$

The directions of the transitions of the TAs for the first input bit, i.e., $x_1/\neg x_1$, when the TAs for the second input bit are frozen, are summarized as follows (based on the subset of the transition diagrams in Appendix 2 of (Jiao et al., 2022)).

**Scenario 1:** Study $TA_1 = I$ and $TA_2 = I$.

**Case 1:** we have
$TA_1 \rightarrow E$
$TA_2 \rightarrow E$
**Case 2:** we have
$TA_1 \rightarrow E$
$TA_2 \rightarrow E$

**Case 3:** we have
$TA_1 \rightarrow E$
$TA_2 \rightarrow E$
**Case 4:** we have
$TA_1 \rightarrow E$
$TA_2 \rightarrow E$

**Scenario 2:** Study $TA_1 = I$ and $TA_2 = E$.

**Case 1:** we have
$TA_1 \rightarrow E$
$TA_2 \rightarrow E$
**Case 2:** we have
$TA_1 \rightarrow E$
$TA_2 \rightarrow E$

**Case 3:** we have
$TA_1 \rightarrow E$
$TA_2 \rightarrow E$
**Case 4:** we have
$TA_1 \rightarrow E$
$TA_2 \rightarrow E$

**Scenario 3:** Study $TA_1 = E$ and $TA_2 = I$.

**Case 1:** we have
$TA_1 \rightarrow I$, or E
$TA_2 \rightarrow E$
**Case 2:** we have
$TA_1 \rightarrow E$
$TA_2 \rightarrow I$

**Case 3:** we have
$TA_1 \rightarrow I$
$TA_2 \rightarrow I$
**Case 4:** we have
$TA_1 \rightarrow E$
$TA_2 \rightarrow E$

**Scenario 4:** Study $TA_1 = E$ and $TA_2 = E$.

**Case 1:** we have
$TA_1 \rightarrow I$, or E
$TA_2 \rightarrow E$
**Case 2:** we have
$TA_1 \rightarrow E$
$TA_2 \rightarrow I$

**Case 3:** we have
$TA_1 \rightarrow E$
$TA_2 \rightarrow E$
**Case 4:** we have
$TA_1 \rightarrow E$
$TA_2 \rightarrow E$

By analyzing the transitions of TAs for the two input bits with samples following Eq. (21), we can conclude that $TA_1 = E$, $TA_2 = I$, $TA_3 = I$, and $TA_4 = E$ is an absorbing state, as the actions of $TA_1$–$TA_4$ reinforce each other to transit to deeper states for the current actions upon various input samples. There are a few other cases in different scenarios that seem to be absorbing, but in fact not. For example, the status $TA_3 = I$ and $TA_4 = E$ seems also absorbing in Scenario 2, Case 3, i.e., when $TA_1 = E$ and $TA_2 = E$ hold. However, to make $TA_1 = E$ and $TA_2 = E$ absorbing, the condition is $TA_3 = I$ and $TA_4 = I$, or $TA_3 = E$ and $TA_4 = E$. Clearly, the status $TA_3 = I$ and $TA_4 = I$ is not absorbing. For $TA_3 = E$ and $TA_4 = E$ to be absorbing, it is required to have $TA_1 = I$ and $TA_2 = I$ to be absorbing, or $TA_1 = I$ and $TA_2 = E$ to be absorbing, which are not true. Therefore, all those absorbing-like states are not absorbing. In fact, when $TA_3 = I$, $TA_4 = E$, $TA_1 = E$, and $TA_2 = E$ hold, the condition $TA_3 = I$, $TA_4 = E$ will reinforce $TA_1$ and $TA_2$

to move towards E, I, which is the absorbing state of the system. Based on the above analysis on the transition directions, we can thus confirm the convergence of TM when training samples from Eq. (21) are given.

Following the same principle, we can also confirm that the TAs will converge to $TA_1 = I$, $TA_2 = E$, $TA_3 = E$, and $TA_4 = I$ when training samples from Eq. (22) are given, according to the proof of Lemma 2 in (Jiao et al., 2022).

## D   APPENDIX: ANALYSIS OF THE TM WITH WRONG TRAINING LABELS

In this appendix, we analyze the transition properties of the TM when training samples contain wrong labels.

There are two types of wrong labels:

- Inputs labeled as 0, which should be 1.
- Inputs labeled as 1, which should be 0.

We begin by examining the first type of wrong label, followed by the second type, and then address the general case.

### D.1   THE AND OPERATOR WITH THE FIRST TYPE OF WRONG LABELS

To formally define training samples with the first type of wrong label, we use the following formulas:

$$P\left(y = 1 | x_1 = 1, x_2 = 1\right) = a, a \in (0, 1), \tag{23}$$
$$P\left(y = 0 | x_1 = 1, x_2 = 1\right) = 1 - a,$$
$$P\left(y = 0 | x_1 = 0, x_2 = 1\right) = 1,$$
$$P\left(y = 0 | x_1 = 1, x_2 = 0\right) = 1,$$
$$P\left(y = 0 | x_1 = 0, x_2 = 0\right) = 1.$$

In this case, the label for training samples representing the intended logic $[x_1 = 1, x_2 = 1]$ is $y = 1$ with probability $a$ and $y = 0$ with probability $1 - a$. In other words, in addition to the training samples detailed in Subsection B, a new training sample will appear to the system, namely $([x_1 = 1, x_2 = 1], y = 0)$. Similar to the noise free studies, we assume the training samples are independently drawn at random, and the above five cases will appear with non-zero probability, which means that all of the five types of samples will appear for infinite number of times given infinite time horizon.

**Lemma 6.** *The TM exhibits recurrence for the training samples defined in Eq. (23).*

**Proof:** To prove this lemma, we analyze the TM's transitions as follows. First, we examine the transitions assuming $u_1 > 0$ and $u_2 > 0$, similar to the analysis in Subsection B, as detailed in Subsection D.1.1. Next, we study the impact of $T$ to determine whether it leads to convergence (absorption), as discussed in Subsection D.1.2.

#### D.1.1   TRANSITION OF TM WITH AND OPERATOR GIVEN $u_1 > 0$ AND $u_2 > 0$

Following the approach in Subsection B, we examine the transitions of $\text{TA}_3$ and $\text{TA}_4$ when the additional training sample $([x_1 = 1, x_2 = 1], y = 0)$ is introduced, considering Cases 1 to 4 as defined in Subsection B. Since $y = 0$ for this sample, only Type II feedback can be triggered to cause transitions. As $\text{TA}_3$ is responsible for the literal $x_2$, which is always 1 for this sample, Type II feedback does not trigger any transitions for $\text{TA}_3$. Therefore, we focus on studying the potential transitions of $\text{TA}_4$ in the four cases defined in Subsection B.1.

In Case 1, where $\text{TA}_1 = E$ and $\text{TA}_2 = I$, the clause value will always be 0 for the training sample because $\neg x_1$ is included in the clause, regardless of the action $\text{TA}_4$ takes. According to the Type II feedback transition table, no transition occurs when $C = 0$, so no transitions are triggered for $\text{TA}_4$. Similarly, in Case 4, where $\text{TA}_1 = I$ and $\text{TA}_2 = I$, the clause value will always be 0 due to the presence of $x_1 \wedge \neg x_1$ in the clause. As a result, there are no transitions for $\text{TA}_4$.

In Case 2, where $\text{TA}_1 = I$ and $\text{TA}_2 = E$, the literal $x_1$ will always appear in the clause. When $\text{TA}_4 = I$, the clause includes the literal $\neg x_2$, which results in a clause value of 0. Therefore, no transition is triggered. However, when $\text{TA}_4 = E$, the literal $x_1$ will always appear in the clause, and the value of $x_2$ is 1, making the clause value 1 regardless of $\text{TA}_3$'s action (whether it includes or excludes $x_2$). According to the Type II feedback table, with the literal value of $\neg x_2$ being 0 and the clause value being 1, the transition for $\text{TA}_4 = E$ is:

Condition: $x_1 = 1, x_2 = 1, y = 0$.
Thus, Type II, $\neg x_2 = 0$,
$C = 1$.

In Case 3, where $TA_1 = E$ and $TA_2 = E$, the clause value is fully determined by $TA_3$ and $TA_4$. When $TA_4$'s action is include, the clause value is 0 for this sample because it includes the literal $\neg x_2$, resulting in no transition for $TA_4$. However, when $TA_4$'s action is to exclude, the clause value is always 1, regardless of $TA_3$'s action. Specifically, when $TA_3$ includes $x_2$, the clause value is 1, as the literal value of $x_2$ is 1. When it is exclude, all literals are excluded and then the clause value becomes 1 by definition. By examining the transitions of $TA_4$, we can summarize the following graph:

Condition: $x_1 = 1, x_2 = 1, y = 0$.
Thus, Type II, $\neg x_2 = 0$,
$C = 1$.

We summarize the directions of the transitions when the new wrongly labeled sample is added, with the newly added actions highlighted in red.

**Scenario 1:** Study $TA_3 = I$ and $TA_4 = I$.

**Case 1**, we have:          **Case 3**, we have:
$TA_3 \Rightarrow E$.          $TA_3 \Rightarrow E$.
$TA_4 \Rightarrow E$.          $TA_4 \Rightarrow E$.
**Case 2**, we have:          **Case 4**, we have:
$TA_3 \Rightarrow E$.          $TA_3 \Rightarrow E$.
$TA_4 \Rightarrow E$.          $TA_4 \Rightarrow E$.

**Scenario 2:** Study $TA_3 = I$ and $TA_4 = E$.

**Case 1**, we have:          **Case 3**, we have:
$TA_3 \Rightarrow E$.          $TA_3 \Rightarrow I$.
$TA_4 \Rightarrow E$ or $I$.          $TA_4 \Rightarrow E$ or $I$.
**Case 2**, we have:          **Case 4**, we have:
$TA_3 \Rightarrow I$.          $TA_3 \Rightarrow E$.
$TA_4 \Rightarrow E$ or I.          $TA_4 \Rightarrow E$.

**Scenario 3:** Study $TA_3 = E$ and $TA_4 = I$.

**Case 1**, we have:          **Case 3**, we have:
$TA_3 \Rightarrow E$ or $I$.          $TA_3 \Rightarrow E$ or $I$.
$TA_4 \Rightarrow E$.          $TA_4 \Rightarrow E$.
**Case 2**, we have:          **Case 4**, we have:
$TA_3 \Rightarrow E$ or $I$.          $TA_3 \Rightarrow E$.
$TA_4 \Rightarrow E$.          $TA_4 \Rightarrow E$.

**Scenario 4:** Study $TA_3 = E$ and $TA_4 = E$.

**Case 1**, we have:          **Case 3**, we have:
$TA_3 \Rightarrow I$ or $E$.          $TA_3 \Rightarrow I$.
$TA_4 \Rightarrow I$ or $E$.          $TA_4 \Rightarrow I$ or $E$.
**Case 2**, we have:          **Case 4**, we have:
$TA_3 \Rightarrow I$.          $TA_3 \Rightarrow E$.
$TA_4 \Rightarrow E$ or I.          $TA_4 \Rightarrow E$.

Clearly, the only absorbing state ($TA_3 = I$ and $TA_4 = E$) becomes recurrent due to the newly added transition (the red $I$ for $TA_4$). As a result, the system is recurrent when $u_1 > 0$ and $u_2 > 0$.

### D.1.2 Transition of TM with AND Operator when $T$ can block Type I feedback

Based on the above analysis, we understand that the system is recurrent when $u_1 > 0$ and $u_2 > 0$. Next, we will examine whether there is any possibility of the system becoming absorbing when $T$ can block Type I feedback.

Clearly, when $T$ clauses learn the intended pattern $\mathbf{X} = [x_1 = 1, x_2 = 1]$, i.e., when $f_{\sum}(\mathcal{C}^i(\mathbf{X})) = T$, $u_1 = 0$ holds, and Type I feedback is blocked. In this situation, only Type II feedback can occur. Due to the presence of the wrong label, i.e., $([x_1 = 1, x_2 = 1], y = 0)$, Type II feedback triggers transitions in the TAs that have already learned the intended logic $(([x_1 = 1, x_2 = 1], y = 1))$. For example, Type II feedback will cause a transition in TAs of a learned clause $C = x_1 \wedge x_2$, making the clause deviate from its learned state (e.g., changing from $x_1 \wedge x_2$ to $x_1 \wedge x_2 \wedge \neg x_2$). Once this happens, $u_1 > 0$ holds, and Type I feedback is triggered by samples of $([x_1 = 1, x_2 = 1], y = 1)$, encouraging TAs in this clause to move back toward the action Exclude. Thus, even when $T$ blocks all Type I feedback samples (setting $u_1 = 0$), the system remains recurrent due to the wrong label and Type II feedback. Notably, no value of $f_{\sum}(\mathcal{C}^i(\mathbf{X}))$ can make both $u_1 = 0$ and $u_2 = 0$ simultaneously[4]. Therefore, Type I and Type II feedback cannot be blocked simultaneously, ensuring the system is recurrent. ∎

### D.2 The AND Operator with the Second Type of Wrong Labels

To properly define the training samples with the second type of wrong label, we employ the following formulas:

$$P(y = 1 | x_1 = 1, x_2 = 1) = 1, \tag{24}$$
$$P(y = 0 | x_1 = 1, x_2 = 0) = a, a \in (0, 1)$$
$$P(y = 1 | x_1 = 1, x_2 = 0) = 1 - a,$$
$$P(y = 0 | x_1 = 0, x_2 = 1) = 1,$$
$$P(y = 0 | x_1 = 0, x_2 = 0) = 1.$$

In this case, clearly, label of the training samples $[x_1 = 1, x_2 = 0]$ are wrongly labeled as 1 with probability $1 - a$. In other words, in addition to the training samples detailed in Subsection B, a new (wrongly labeled) training sample will appear to the system, namely $([x_1 = 1, x_2 = 0], y = 1)$.

**Lemma 7.** *The TM is recurrent for the training samples given by Eq. (24).*

**Proof:** Similar to the proof of Lemma 6, we first consider the transitions of TM with $u_1 > 0$ and $u_2 > 0$, and then examine the impact of $T$ for the system transition.

Clearly, when $u_1 > 0$, $u_2 > 0$ holds, there is a non-zero probability that training sample $([x_1 = 1, x_2 = 0], y = 1)$ will appear to the system. The appearance of this sample will involve transition of $TA_3$ moving from action Include toward Exclude, as shown in Fig. 3, making the system recurrent.

Now we study if the functionality of $T$ can offer system absorption. When $T$ clauses learn the intended pattern $\mathbf{X} = [x_1 = 1, x_2 = 1]$, i.e., $f_{\sum}(\mathcal{C}^i(\mathbf{X})) = T$, $u_1 = 0$ holds, and thus Type I feedback is blocked for this training sample. In this situation, the TM can only see the training samples following:

$$P(y = 0 | x_1 = 1, x_2 = 0) = a, a \in (0, 1) \tag{25}$$
$$P(y = 1 | x_1 = 1, x_2 = 0) = 1 - a,$$
$$P(y = 0 | x_1 = 0, x_2 = 1) = 1,$$
$$P(y = 0 | x_1 = 0, x_2 = 0) = 1.$$

Following the same concept as the proof of Lemma 6, we can conclude that the TM is recurrent for the samples in Eq. (25). Clearly, the system is recurrent, regardless of the value of $u_1$. Therefore, we can conclude that the TM is recurrent for the training samples described in Eq.(24).

---

[4]In this study, we focus only on positive polarity thus $u_2 > 0$ always holds. When negative polarity is enabled (i.e., when a set of clauses learns sub-patterns with label $y = 0$), $u_2$ becomes 0 when $T$ clauses learn a sample with $y = 0$. However, it remains true that no value of $f_{\sum}(\mathcal{C}^i(\mathbf{X}))$ can make both $u_1$ and $u_2$ equal to 0 simultaneously.

Following the same principle, we can also prove that the TM is recurrent when other training samples, i.e., $[x_1 = 0, x_2 = 1]$, and $[x_1 = 0, x_2 = 0]$, or their combinations, have wrong labels. We thus can conclude that the TM is recurrent for the second type of wrong labels. ∎

So far, we have proven that the TM is recurrent when only one type of wrong label exists for the AND operator. It is straightforward to conclude that the TM remains recurrent when both types of wrong labels are present. The key reason is that adding both types of wrong labels does not eliminate any transitions between system states in recurrent systems. Therefore, the TM is recurrent for training samples with general wrong labels for the AND operator. Using the same reasoning, we can extend this conclusion to the XOR and OR operators. Thus, the following theorem holds.

**Theorem 6.** *The TM is recurrent given training samples with wrong labels for the AND, OR, and XOR operators.*

**Remark 5.** *The primary reason for the recurrent behavior of the TM when wrong labels are present is the introduction of statistically conflicting labels for the same input samples. These inconsistency causes the TAs within a clause to learn conflicting outcomes for the same input due to the corresponding Type I and Type II feedback for label 1 and 0 respectively. When a clause learns to evaluate an input as 1 based on Type I feedback, samples with a label of 0 for the same input prompt it to learn the input as 0 during Type II feedback. This conflict in labels confuses the TM, leading to back-and-forth learning.*

**Remark 6.** *Note that although wrong labels will make the TM not converge (not absorbing with 100% accuracy for the intended logic), via simulations, we can still find that the TM can learn the operators efficiently, which is to be demonstrated in Appendix G. Interestingly, when the probability of the second type of wrong label is large, TM will consider it as a sub-pattern, and learn it, which aligns with the nature of learning.*

# E  APPENDIX: ANALYSIS OF THE TM WITH IRRELEVANT INPUT VARIABLES

In this appendix, we examine the impact of random irrelevant input variables on the TM. An irrelevant variable refers to an input bit with a random value that does not affect the classification result. For instance, in the AND operator, a third input bit, $x_3$, may appear in the training sample with random 1 and 0 values, but its value does not influence the output of the AND operator. In other words, the output is entirely determined by the values of $x_1$ and $x_2$. Formally, we have:

$$P\left(y = 1 | x_1 = 1, x_2 = 1, x_3 = 0 \: or \: 1\right) = 1, \tag{26}$$
$$P\left(y = 0 | x_1 = 1, x_2 = 0, x_3 = 0 \: or \: 1\right) = 1,$$
$$P\left(y = 0 | x_1 = 0, x_2 = 1, x_3 = 0 \: or \: 1\right) = 1,$$
$$P\left(y = 0 | x_1 = 0, x_2 = 0, x_3 = 0 \: or \: 1\right) = 1.$$

Here $x_3 = 0 \: or \: 1$ means $P(x_3 = 0) = a$, $P(x_3 = 1) = 1 - a$, $a \in (0, 1)$. We assume the training samples are independently drawn at random, and the above four cases will appear with non-zero probability, which means that all of the four types of samples will appear for infinite number of times given infinite time horizon.

## E.1  CONVERGENCE ANALYSIS OF THE AND OPERATOR WITH AN IRRELEVANT VARIABLE

**Theorem 7.** *The clauses in a TM can almost surely learn the AND logic given training samples in Eq. (26) in infinite time, when $T \leq m$.*

**Proof:** The proof of Theorem 7 consists of two steps: (1) Identifying a set of absorbing conditions and confirming that the TM, when in these conditions, satisfies the requirements of the AND operator. (2) Demonstrating that any state of the TM that deviates from the conditions defined in step (1) is not absorbing.

The TM will be absorbed when the following conditions fulfill:

1. Condition to block Type I feedback: For any input sample $\mathbf{X} = [x_1 = 1, x_2 = 1, x_3]$, regardless of whether $x_3 = 1$ or 0, the TM has at least $T$ clauses that output 1.

2. Conditions to guarantee no transitions upon Type II feedback:

   (a) When $x_3$ or $\neg x_3$ appears in a clause in the TM: The literals that are included in the clause for the first two input variables must result in a clause value of 0 for the input samples $\mathbf{X} = [x_1 = 0, x_2 = 1, x_3]$, $\mathbf{X} = [x_1 = 1, x_2 = 0, x_3]$ and $\mathbf{X} = [x_1 = 0, x_2 = 0, x_3]$. This ensures that $C = 0$ for these input samples, regardless of the value of $x_3$, thereby preventing transitions caused by any Type II feedback. The portion of the clause involving the first two input variables can be, e.g., $x_1 \wedge x_2$ or $x_1 \wedge \neg x_1 \wedge x_2$, while the overall clauses can be, e.g., $C = x_1 \wedge x_2 \wedge x_3$, or $C = x_1 \wedge \neg x_1 \wedge x_2 \wedge \neg x_3$, as long as the resulted clause value is 0 for those input samples.

   (b) When $x_3$ or $\neg x_3$ does NOT appear in a clause in the TM: There is no clause that is in the form of $C = x_1$, $C = x_2$, $C = x_1 \wedge \neg x_2$, $C = \neg x_1 \wedge x_2$, $C = \neg x_1$, $C = \neg x_2$, or $C = \neg x_1 \wedge \neg x_2$.

Clearly, when the above conditions fulfill, the system is in absorption because no feedback appears to the system. Additionally, this absorbing state follows AND operator. Based on the statement of the condition to block Type I feedback, there are at least $T$ clauses that output 1 for input sample $\mathbf{X} = [x_1 = 1, x_2 = 1, x_3]$, regardless $x_3 = 1$ or 0. Studying the conditions for Type II feedback, we can conclude that the clause outputs 0 for all input samples $\mathbf{X} = [x_1 = 1, x_2 = 0, x_3]$, $\mathbf{X} = [x_1 = 0, x_2 = 1, x_3]$, or $\mathbf{X} = [x_1 = 0, x_2 = 0, x_3]$. We can then setup the $Th = T$ to confirm the AND logic.

The next step is to show that any state of the TM deviating from the above conditions is not absorbing. To demonstrate this, we can simply confirm that transitions, which might change the current actions of the TAs, will occur due to updates from Type I or Type II feedback.

When literal $x_3$ or literal $\neg x_3$ is included as a part of the clause, there is non-zero probability for $C = 0$ due to the randomness of input variable $x_3$. As a result, Type I Feedback will encourage

the TA for the included literal $x_3$ or $\neg x_3$ to move away from its current action, thus preventing the system from becoming absorbing.

For the case where literal $x_3$ or literal $\neg x_3$ is not included in the clause, the system operates purely based on the first two input variables, namely $x_1$ and $x_2$. According our previous analysis for the noise free AND case (Theorem 1), there is only one absorbing status, which is $C = x_1 \wedge x_2$. However, this absorbing state disappears because Type I feedback will encourage the excluded literal $x_3$ to be included when $x_3 = 1$, and similarly encourage the excluded literal $\neg x_3$ to be included when $x_3 = 0$. Once either $x_3$ or $\neg x_3$ is included, the analysis in the previous paragraph applies, and thus the system is not absorbing.

From the above discussion, it is clear that Type I feedback is the key driver of action changes in non-absorbing cases. If Type I feedback is not blocked, the system cannot reach an absorbing state. Therefore, blocking Type I feedback is critical for achieving convergence. The condition $T < m$ is to guarantee that $T$ should not be greater than the total number of clauses, making it feasible to block Type I feedback. ∎

**Remark 7.** *Due to the existence of the noisy input variable $x_3$, the system requires the functionality of $T$ to block Type I feedback in order to converge. This contrasts with the noise-free case, where the TM will almost surely converge to the AND operator even when Type I feedback is consistently present ($u_1 > 0$).*

E.2 CONVERGENCE ANALYSIS OF THE OR OPERATOR WITH AN IRRELEVANT VARIABLE

For the OR case, we have

$$
\begin{aligned}
P\left(y = 1 | x_1 = 1, x_2 = 1, x_3 = 0 \text{ or } 1\right) &= 1, \\
P\left(y = 1 | x_1 = 1, x_2 = 0, x_3 = 0 \text{ or } 1\right) &= 1, \\
P\left(y = 1 | x_1 = 0, x_2 = 1, x_3 = 0 \text{ or } 1\right) &= 1, \\
P\left(y = 0 | x_1 = 0, x_2 = 0, x_3 = 0 \text{ or } 1\right) &= 1.
\end{aligned}
\tag{27}
$$

**Theorem 8.** *The clauses in a TM can almost surely learn the OR logic given training samples in Eq. (27) in infinite time, when $T \leq \lfloor m/2 \rfloor$.*

**Proof:** The proof of Theorem 8 follows a similar structure to that of the AND case and involves two steps: (1) Identifying a set of absorbing conditions and verifying that, under these conditions, the TM satisfies the requirements of the OR operator. (2) demonstrating that any state of the TM deviating from these conditions is not absorbing.

1. Condition to block Type I feedback: For any input sample $\mathbf{X} = [x_1 = 1, x_2 = 1, x_3]$, $\mathbf{X} = [x_1 = 1, x_2 = 0, x_3]$, and $\mathbf{X} = [x_1 = 0, x_2 = 1, x_3]$ regardless of whether $x_3 = 1$ or $0$, the TM has at least $T$ clauses that output 1.

2. Conditions to guarantee no transitions upon Type II feedback:

   (a) When $x_3$ or $\neg x_3$ appears in a clause in the TM: The literals included in the clause for the first two input variables must ensure a clause value of 0 for the input samples $\mathbf{X} = [x_1 = 0, x_2 = 0, x_3]$. This is to guarantee that $C = 0$ for those input samples, irrespective of the value of $x_3$, thereby preventing any transitions caused by Type II feedback. The portion of the clause involving the first two input variables can take the form such as $x_1, x_1 \wedge \neg x_2, x_1 \wedge x_2, x_1 \wedge \neg x_1 \wedge x_2$. Correspondingly, the overall clauses can take the form such as $C = x_1 \wedge \neg x_3$, $C = x_1 \wedge \neg x_2 \wedge x_3$, $C = x_1 \wedge x_2 \wedge x_3$, or $C = x_1 \wedge \neg x_1 \wedge x_2 \wedge \neg x_3$, as long as the resulted clause value is 0 for those input samples.

   (b) When $x_3$ or $\neg x_3$ does not appear in a clause in the TM: There is no clause with literal(s) in only negated form, such as $C = \neg x_1$, $C = \neg x_2$, or $C = \neg x_1 \wedge \neg x_2$.

Clearly, when the above conditions fulfill, the system is absorbing because no feedback appears to the system. Additionally, this absorbing state adheres to the OR operator. Based on the condition required to block Type I feedback, there are at least $T$ clauses that output 1 for input sample $\mathbf{X} = [x_1 = 1, x_2 = 1, x_3]$, $\mathbf{X} = [x_1 = 1, x_2 = 0, x_3]$, or $\mathbf{X} = [x_1 = 0, x_2 = 1, x_3]$ regardless of whether

$x_3 = 1$ or 0. Analyzing the conditions for Type II feedback, we find that the clause outputs 0 for all input samples $\mathbf{X} = [x_1 = 0, x_2 = 0, x_3]$. We can then setup the $Th = T$ to confirm the OR logic.

The next step is to demonstrate that any state of the TM that deviates from the above conditions outlined above is not absorbing. To do this, we can confirm that transitions which may alter the current actions of the TAs will occur due to updates from Type I and Type II feedback.

When literal $x_3$ or literal $\neg x_3$ is included in the clause, there is non-zero probability for $C = 0$ due to the randomness of input variable $x_3$. In this case, Type I Feedback will move the included literal $x_3$ or $\neg x_3$ towards action Exclude, preventing the system from being absorbing.

For the case where literal $x_3$ or literal $\neg x_3$ is not included as a part of the clause, the system operates purely based on the first two input variables, namely $x_1$ and $x_2$. Based on our previous analysis for the noise free OR case shown in Lemma 2, the system is recurrent. This recurrent behavior will eventually lead the system to a state where the excluded literal, either $x_3$ or $\neg x_3$, is encouraged to be included. For example, if the TM has a clause $C = x_1 \wedge x_2$, upon a training sample $\mathbf{X} = [x_1 = 1, x_2 = 1, x_3 = 0]$, the Type I feedback will encourage the excluded literal $\neg x_3$ to be included. Once one of the excluded literal, $x_3$ or $\neg x_3$, is included, the analysis in the previous paragraph applies, meaning the system is not absorbing.

Clearly, if Type I feedback is not blocked, the system will not be absorbing. As blocking Type I feedback is critical, condition $T \leq \lfloor m/2 \rfloor$ is necessary, refer to Lemma 4. ∎

When $T$ clauses have learned the intended sub-patterns of OR operation, the Type I feedback will be blocked. At the same time, Type II feedback will eliminate all clauses that output 1 for input sample following $\mathbf{X} = [x_1 = 0, x_2 = 0, x_3]$, removing false positives. At this point, the system has converged. The presence of $x_3$ does not change the convergence feature, but it adds more dynamics to the TM.

### E.3 Convergence Analysis of the XOR Operator with an Irrelevant Variable

**Theorem 9.** *The clauses in a TM can almost surely learn the XOR logic given training samples in Eq. (28) in infinite time, when $T \leq \lfloor m/2 \rfloor$.*

$$
\begin{aligned}
P\left(y = 0 | x_1 = 1, x_2 = 1, x_3 = 0 \text{ or } 1\right) &= 1, \\
P\left(y = 1 | x_1 = 1, x_2 = 0, x_3 = 0 \text{ or } 1\right) &= 1, \\
P\left(y = 1 | x_1 = 0, x_2 = 1, x_3 = 0 \text{ or } 1\right) &= 1, \\
P\left(y = 0 | x_1 = 0, x_2 = 0, x_3 = 0 \text{ or } 1\right) &= 1.
\end{aligned}
\tag{28}
$$

The proof for XOR follows the same principles as the AND and OR cases, and therefore, we do not present it explicitly here.

### E.4 Convergence Analysis of the Operators with Multiple Irrelevant Variables

In the previous subsections, we demonstrated that if a single irrelevant bit is present in the training samples, the system will almost surely converge to the intended operators. This conclusion can be readily extended to scenarios involving multiple irrelevant variables. Here, "multiple irrelevant variables" refers to the presence of additional variables, beyond $x_3$, in the training samples that do not contribute to the classification.

**Theorem 10.** *The clauses in a TM can almost surely learn the 2-bit AND logic given training samples with $k$ irrelevant input variables in infinite time, $0 < k < \infty$, when $T \leq m$.*

**Theorem 11.** *The clauses in a TM can almost surely learn the 2-bit XOR and OR logic given training samples with $k$ irrelevant input variables in infinite time, $0 < k < \infty$, when $T \leq \lfloor m/2 \rfloor$.*

**Proof:** The proofs of Theorems 10 and 11 are straightforward. We must verify whether the conditions for blocking Type I feedback and resulting no transitions upon Type II feedback remain valid when multiple irrelevant variables are present. In addition, we show that the system is still recurrent when those conditions do not fulfill.

The condition for blocking Type I feedback remains valid because Type I feedback is only determined by the first two input bits and is not a function of the irrelevant variables. For Type II feedback, its effect depends on whether the literals for the irrelevant input variables are present in the clause. In cases where the literals of the irrelevant bits are not included in the clause, the analysis holds, as those literals are absent. When the literals of the irrelevant bits are included, their number does not impact the analysis. This is because the clause value is entirely determined by the first two bits, and the clause value remains $C = 0$, regardless of the number of irrelevant variables.

The system is recurrent when these conditions for absorbing do not fulfill. We have shown that the system is recurrent when one irrelevant label $x_3$ exists, and the same analysis applies for the transitions with more irrelevant variables. In addition, any extra irrelevant variable does not eliminate any transitions in the original system before it is added. Therefore, the system is still recurrent before the absorbing conditions fulfill. We constrain the number of irrelevant variables to an infinite number to avoid the analysis on a system with infinite states. ∎

# F    Appendix: Experiment Results with Noise-Free Training Samples

To validate the theoretical analyses, we here present the experiment results[5] for both the AND and the OR operators.

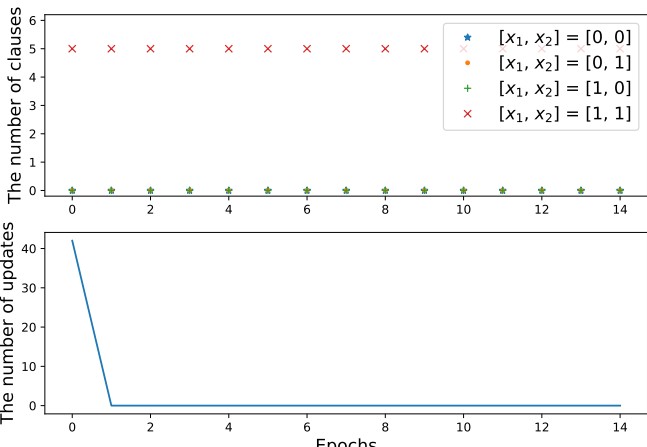

Figure 8: The convergence of a TM with 7 clauses when $T = 5$ for the AND operator.

Figure 8 shows the convergence of TM for the AND operator when $m = 7$, $T = 5$, $s = 4$, and $N = 50$. More specifically, we plot the number of clauses that learn the AND operator, namely, $x_1 = x_2 = 1$, and the number of system updates as a function of epochs. From these figures, we can clearly observe that after a few epochs, the TM has 5 clauses that learn the AND operator and then the system stops updating because no update is triggered anymore. Note that if we control $T$ so that $u_1 > 0$ always holds, all clauses will converge to the AND operator, which has been validated via experiments. These observations confirm Theorem 1. Although the theorem says it may require infinite time in principle, the actual convergence can be much faster.

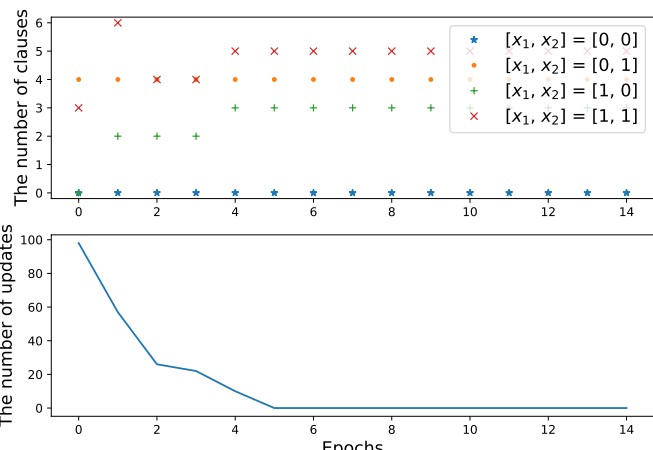

Figure 9: The convergence of a TM with 7 clauses when $T = 3$ for the OR operator.

In Fig. 9, we illustrate the number of clauses in distinct sub-patterns when we employ $m = 7$, $T = 3$, $s = 4$, and $N = 50$ for the OR operator. Based on the analytical result, i.e., Theorem 2, the system will be absorbed, where each sub-pattern will have at least 3 clauses and no update will happen afterwards. From the figure, we can clearly observe that after a few epochs, the system

---

[5]The code for validating the convergence of AND and OR operators can be found at `https://github.com/JaneGlim/Convergence-of-Tsetline-Machine-for-the-AND-OR-operators`.

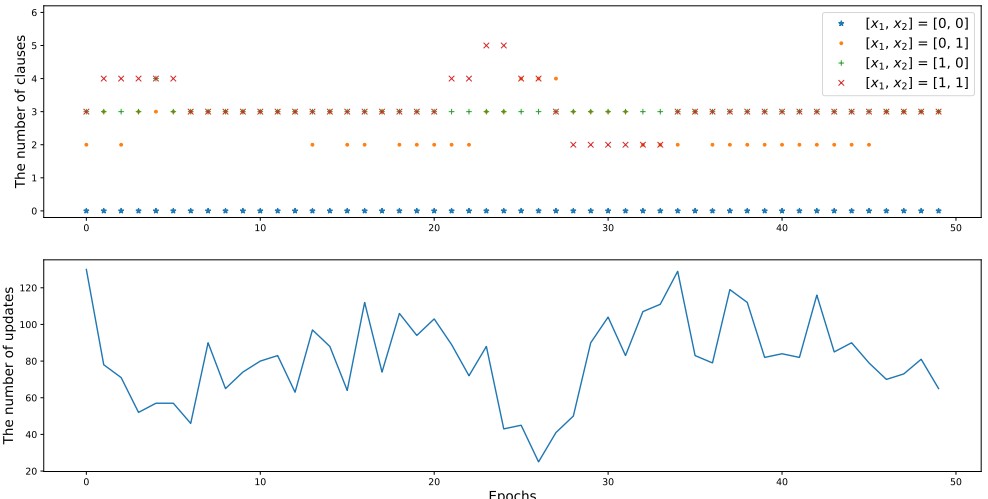

Figure 10: The behavior of a TM with 7 clauses when $T = 4$ for the OR operator.

becomes indeed absorbed as no updates are observed. When absorbed, two intended sub-patterns have 3 clauses individually, one intended sub-pattern has 4 clauses, while the unintended sub-pattern has 0 clause, which coincides with the theorem. After checking the converged actions of the TAs, we find the list of the converged clauses: $C_1 = x_1$, $C_2 = x_1$, $C_3 = x_2$, $C_4 = x_1 \wedge \neg x_2$, $C_5 = x_1 \wedge x_2$, $C_6 = x_2$, and $C_7 = \neg x_1 \wedge x_2$, which explained the number of the converged clauses in different sub-patterns shown in the figure. Clearly, in this example, some clauses, such as $C_1$ and $C_3$, can cover multiple sub-patterns. This indicates that in real world applications, if distinct sub-patterns have certain bits in common, which can be used to differentiate it from other classes, it is possible for TM to learn those bits as jointly features, confirming the efficiency of the TM.

Note that there are many other possible absorbing states that are different from the shown example, which have been observed when we run multiple instances of the experiments. As long as at least $T$ clauses can cover each intended sub-pattern in the OR operator, the system converges.

In Fig. 10, the configuration is identical to that in Fig. 9 other than $T = 4$. In this case, as stated in Remark 2, the system will not become absorbing, but will still cover the intended sub-patterns with high probability. From this figure, we can observe that at least two clauses are able to cover each intended sub-pattern and the unintended sub-pattern has zero clause. At the same time, the TAs will update their states along epochs, which can be seen in the bottom figure. It is worth mentioning that we have occasionally observed in other rounds of experiments, that one sub-pattern is covered by only 1 clause. In this case, it is still possible to set up $Th \geq 1$ to have successful classification. Nevertheless, there is no guarantee that at least one clause will follow each intended sub-pattern in this configuration.

## G  APPENDIX: EXPERIMENT RESULTS WITH NOISY TRAINING SAMPLES

This Appendix presents the experimental results for the operators under noisy conditions. First, we show the results when incorrect labels are present, followed by the results involving irrelevant variables. The final subsection addresses a case where both incorrect labels and irrelevant variables are present.

### G.1  EXPERIMENT RESULTS FOR WRONG LABELS

To evaluate the performance of the TM when exposed to mislabeled samples, we introduced incorrectly labeled data into the system. The key observation is that the TM does not converge to the intended logic, meaning it does not absorb into a state where the correct logic is consistently represented. However, with carefully chosen hyperparameters, the TM can still learn the intended logic with high probability.

To demonstrate the TM's behavior, we first conduct experiments on the OR operator, which satisfies the following equation:

$$P\left(y = 1 | x_1 = 1, x_2 = 1\right) = 90\%,\tag{29}$$
$$P\left(y = 1 | x_1 = 1, x_2 = 0\right) = 90\%,$$
$$P\left(y = 1 | x_1 = 1, x_2 = 0\right) = 90\%,$$
$$P\left(y = 0 | x_1 = 0, x_2 = 0\right) = 1.$$

In this scenario, 10% of the input samples that should be labeled as 1 were incorrectly labeled as 0. To train the TM and evaluate its performance, we used the following hyperparameters: $T = 4$, $Th = 2$, $s = 3$, $m = 7$, and $N = 100$. Fig. 11 shows the number of updates and the number of clauses that learn distinct sub-patterns, as a function of epochs. As shown in Fig. 11, the number of updates is big, and thus the system did not converge. Nevertheless, when examining the number of clauses associated with each sub-pattern, we observed that each sub-pattern was covered by at least two clauses, ensuring that the OR operator remained valid. Similar results were observed in experiments conducted on the AND and XOR operators.

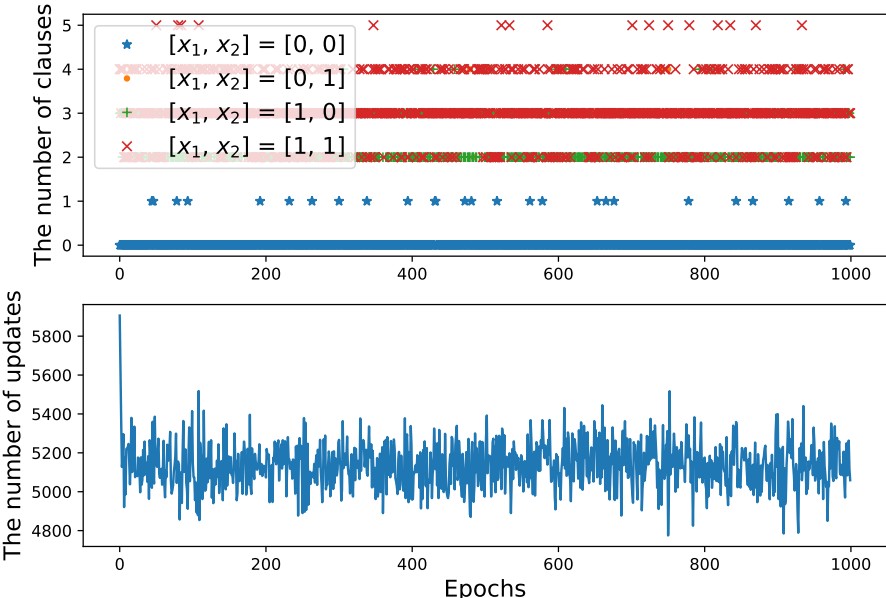

Figure 11: The behavior of TM when $m = 7$, $T = 4$ for the OR operator with wrong training labels.

Interestingly, when the proportion of mislabeled samples increases to an extreme level, where inputs that should be labeled as 0 are instead labeled as 1, the TM begins to treat the noise as a sub-pattern. For instance, consider the AND operator with input $\mathbf{X} = [x_1 = 0, x_2 = 1]$, which is mislabeled as 1 in 90% of the cases, as shown in Eq. (30). Using the hyperparameters $T = 3$, $s = 3.0$, $m = 7$, and $N = 100$, we observed from experiments that the TM generates three clauses with an output of 1 for $\mathbf{X} = [x_1 = 0, x_2 = 1]$ and another three clauses with an output of 1 for $\mathbf{X} = [x_1 = 1, x_2 = 1]$. This behavior indicates that the TM has incorporated the noise as a learned sub-pattern. Such outcomes align with the TM's underlying principle of learning, where it identifies and models sub-patterns associated with the label 1.

$$P\left(y = 1 | x_1 = 1, x_2 = 1\right) = 1,$$
$$P\left(y = 0 | x_1 = 1, x_2 = 0\right) = 1,$$
$$P\left(y = 1 | x_1 = 0, x_2 = 1\right) = 90\%,$$
$$P\left(y = 0 | x_1 = 0, x_2 = 0\right) = 1. \tag{30}$$

### G.2 Experiment Results for Irrelevant Variable

To confirm the convergence property of TM with irrelevant variable, we setup the experiments for the AND, OR, and XOR operators when one irrelevant variable, namely, $x_3$, exists. The probability of $x_3$ being one in the training and testing samples is 50%.

For the AND operator, we use the hyperparameters $m = 5$, $T = 2$, $s = 3$, $Th = 2$, and $N = 100$. Fig. 12 illustrates the convergence of TM for the AND operator in the presence of an irrelevant bit. The results confirm that the TM can correctly learn the AND operator without uncertainty, validating the correctness of Theorem 7.

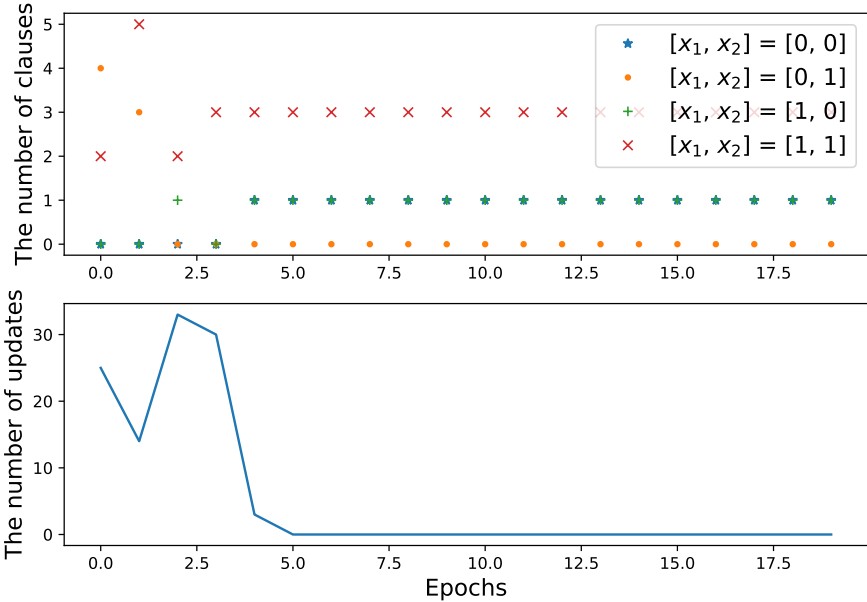

Figure 12: Convergence of TM when $m = 5$, $T = 2$ for the AND operator with an irrelevant label.

Interestingly, upon convergence, the form of the included literals varies. For instance, with the aforementioned hyperparameters, we observe that the converged TM includes two clauses of the form $x_1 \wedge x_2 \wedge x_3$ and another two clauses of the form $x_1 \wedge x_2 \wedge \neg x_3$. This suggests that, instead of excluding the irrelevant bit $x_3$, the TM includes at least $T$ clauses containing $x_3$ and at least $T$ clauses containing $\neg x_3$, which ensures correct classification regardless of the value of $x_3$. However, when the hyperparameters are set to $m = 1$, $T = 1$, $s = 3$, $Th = 1$, and $N = 100$, where only a

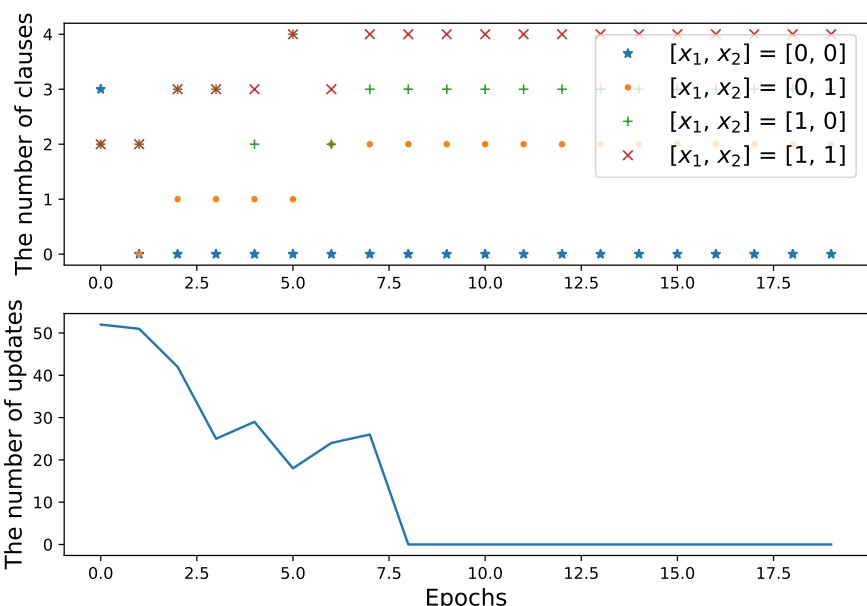

Figure 13: Convergence of TM when $m = 5, T = 2$ for the OR operator with an irrelevant label.

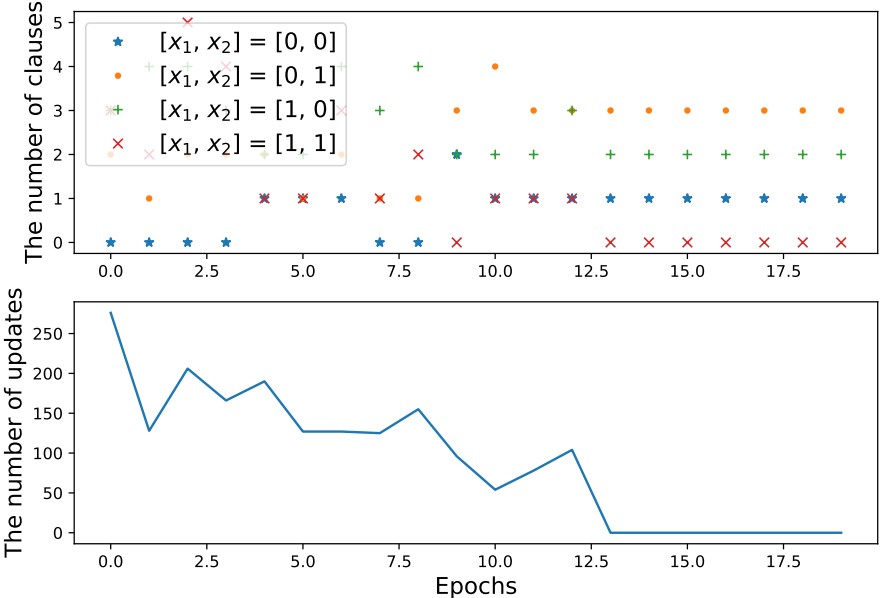

Figure 14: Convergence of TM when $m = 7, T = 2$ for the XOR operator with an irrelevant label.

single clause exists in the TM, the converged clause takes the form $x_1 \wedge x_2$, excluding the literals $x_3$ and $\neg x_3$.

As $T$ increases ($T > m/2$), we observe that convergence becomes challenging. This difficulty arises because the TM cannot simultaneously learn $T$ clauses containing $x_3$ and another $T$ clauses

containing $\neg x_3$. In such cases, the TM must rely on $T$ clauses in the form $x_1 \wedge x_2$ to achieve convergence, which can be particularly demanding.

For the OR operator, we use the hyperparameters $m = 5$, $T = 2$, $s = 3$, $Th = 2$, and $N = 100$. Figure 13 illustrates the convergence of the TM for the OR operator in the presence of an irrelevant bit. The results confirm that the TM successfully learns the OR operator without ambiguity, validating the correctness of Theorem 8.

From the experimental results, we also observe that there are multiple possible absorbing states, as long as the absorbing conditions are satisfied. Additionally, the TM is capable of presenting two sub-patterns simultaneously. Depending on the hyperparameter configuration, $x_3$ and $\neg x_3$ may be included in the clauses, provided that $T$ clauses can align with each intended sub-pattern, which ensures correct classification regardless of the value of $x_3$.

We have also studied the XOR operator. The convergence instance is shown in Fig. 14, confirming Theorem 9. Here we use $m = 7$, $T = 2$, $s = 3$, $Th = 2$.

### G.3 EXPERIMENT RESULTS FOR BOTH WRONG LABELS AND IRRELEVANT VARIABLES

In this experiment, we assess the performance of the TM in the presence of both mislabeled training data and irrelevant variables. Specifically, we evaluate the TM's ability to learn the XOR operator when 40% of the samples are incorrectly labeled, and 10 irrelevant variables are added. The input comprises 12 bits, with only the first two bits determining the output based on the XOR logic. The hyperparameters are configured as follows: $T = 15$, $s = 3.9$, $c = 20$, and $N = 100$ with polarity enabled. Experimental results reveal that the TM successfully learns the XOR operator in 99% of 200 independent runs. These findings demonstrate the robustness of the TM training in noisy environments.

In another experiment, we configured the TM to learn a noisy XOR function with 2 useful input bits and 18 irrelevant input bits (hyper parameters: $N = 128$, $m = 20$, $T = 10$, $s = 3$, label noise 10%). Remarkably, the TM was still able to learn the XOR operator with 100% accuracy using just 5000 training samples. If all possible input variable combinations were required in the training samples, we would require $2^{20} = 1048576$ samples. Clearly, the TM does not rely on the entire exponential input space to learn effectively.

