# OpenReview forum: "On the Convergence of Tsetlin Machines for the AND and the OR Operators"
_ICLR.cc/2025/Conference — Submitted to ICLR 2025_

### Official Review · Reviewer_M3eM · 2024-10-29

**Soundness:** 2
**Presentation:** 1
**Contribution:** 1
**Rating:** 3
**Confidence:** 3

**Summary:**

This paper examines Tsetlin Machines (TMs), a machine learning architecture characterized by notable attributes such as transparent inference and a hardware-friendly design. While recent literature has established convergence properties for the identity, negation, and exclusive-or operators, this study completes the analysis by establishing the convergence of the “AND” and “OR” operators.

**Strengths:**

The authors demonstrate that TMs can almost surely learn the AND and OR relations with the simplest structures, under noise-free conditions.

**Weaknesses:**

**Audience:**
The paper primarily focuses on Tsetlin Machines, particularly their capability to learn logical concepts such as AND and OR in a noise-free environment. However, it does not address other potential approaches that can also learn these simple concepts under similar conditions. Consequently, the paper seems tailored to a niche group of researchers invested in Tsetlin Machines, raising doubts about its relevance to the broader ICLR community.

**Significance:**
The introduction highlights that the main motivation for studying Tsetlin Machines is their interpretability, as their inference is explainable and their learning process is transparent. A crucial question arises: "Are there other interpretable models and transparent algorithms capable of learning logical concepts under noise-free conditions?" The clear answer is yes. Noise-free concept learning is merely equivalent  to "proper and realizable PAC learning," which has been thoroughly explored since the 1980s. Notably, when the target function is a conjunction, disjunction, or exclusive disjunction of literals over two features, the class of 2-DNF formulas covers all these target concepts. Additionally, 2-DNF is both properly and efficiently PAC learnable in the realizable setting, with a sample complexity that is logarithmic in the input dimension - see e.g. landmark papers such as Valiant (1984) and Haussler (1988), and Shalev-Shwartz and Ben-David’s book (2014). Finally, set cover algorithms for learning disjunctions, and more generally k-DNF, are entirely transparent (see e.g. Marchand and Shawe-Taylor, JMRL 2002).

Even if we focus solely on the results concerning Tsetlin Machines, it seems that Jiao et al. (2022) have already established a foundation for learning binary relations, particularly the XOR operation. As a result, the current study, which investigates the convergence of other binary relations (AND, OR), appears to have limited significance, despite some technical differences. Additionally, since the learning process assumes ideal, noise-free conditions without irrelevant variables, this contribution is relatively minor.

**Clarity:**
The paper presents significant readability challenges due to issues with formatting and punctuation. Specifically, citations should be enclosed in brackets, and punctuation should be used sparingly. For instance, the sentence “In TM, after training, a certain clause, in propositional logic, is a conjunction of literals [...]” (Line 32) notably impacts the paper's clarity.

Additionally, for the ICLR community, it is crucial to provide a clear background on these architectures. Unfortunately, Section 2 falls short in this respect. Key concepts such as Tsetlin Automata (TA) and the notion of a team are mentioned without adequate explanation. Actually, I had to consult the papers by Abeyrathna et al. (2021) and Jiao et al. (2022) to fully understand these concepts.

**Questions:**

In light of the above comments:
* Can the convergence proofs for AND and OR be extended to noisy/agnostic settings?
* Alternatively, do these convergence proofs still hold in the presence of many irrelevant variables?

---

> ### Author Response · Authors · 2024-11-20
>
> Thank you for your questions and valuable feedback.
>
> TM is a symbolic AI algorithm with the potential to be structured into a deep connection architecture similar to neural networks. This novel algorithm design gives it the potential to tackle complex machine learning tasks while maintaining interpretability. TM is entirely based on logical operations, making it hardware-friendly and relatively low in power consumption. Like any new algorithm, it needs time to gain widespread acceptance. Although it has not yet reached the level of popularity that deep neural networks enjoy, its following is gradually growing. We believe that publishing at prestigious conferences like ICLR will bring greater visibility to TM, encouraging its application across various fields. We also believe that the ICLR audience will find it interesting.
>
> We completely agree with your statement in the Weakness-Significance section and greatly appreciate your insightful feedback. However, the aim of this paper is to prove that TM is capable of learning the patterns in the AND/OR cases. This is important as TM is a completely novel design of machine learning, hence study on its performance in solving basic classification problems will prove its effectiveness and provide theoretical insights in understanding it fundamentally.
>
> TM is not only capable of learning logic concepts, it has been applied to complex problems. We list here a few relevant research papers:
> - Yadav, R., et al. "Human-level interpretable learning for aspect-based sentiment analysis." AAAI2021.
> - Yadav, R., et al. "Robust interpretable text classification against spurious correlations using AND-rules with negation." IJCAI2022.
> - Seraj, R., et al.  "Tsetlin machine for solving contextual bandit problems." Neurips2022.
> - Maheshwari, S. et al., "REDRESS: Generating Compressed Models for Edge Inference Using Tsetlin Machines," TPAMI2023
>
> TM is also utilized in industrial and public sector applications:
> - Using Tsetlin machine for GPS jamming detection by NKOM (Norwegian communication authority): https://nkom.no/aktuelt/nkom-tester-norsk-ki-modell-i-jakten-pa-gps-forstyrrelser
> - Developing sustainable AI by the company Literal Lab: https://www.forbes.com/sites/charlestowersclark/2024/09/20/unlocking-sustainable-ai-the-game-changing-tsetlin-machine-approach/
>
> Since XOR can be derived from AND, OR and NOT, it might seem that once XOR is proven, proving AND and OR would become trivial—this was even the initial thought of the authors. However, that is not the case. The reasons are:
> Firstly, the learning process of the TM is direct, not derived. For example, instead of using a complicated combination of AND, OR and NOT operators, the TM uses two distinct clauses $\neg x_1 \wedge x_2$ and $x_1 \wedge \neg x_2$ to directly represent the two sub-patterns of XOR (01=>1, 10=>1).
> Secondly, the approaches for the TM to converge to different operators are quite different. In the XOR case, there are two sub-patterns, and they are mutually exclusive. In the OR case, there are three sub-patterns, which are not mutually exclusive. The clauses learned by the TM to represent sub-patterns in OR can be $\neg x_1 \wedge x_2$, $x_1 \wedge \neg x_2$, $x_1 \wedge x_2$, $x_1$, and $x_2$. The proof of XOR reveals how a TM learns mutually-exclusive sub-patterns and how the hyper-parameter T functions in the learning process, while the proof of OR shows how non-mutually-exclusive sub-patterns can be learned jointly by the TM.
>
> For the above reasons, the learning mechanism for these basic operators deserve individual in-depth study.
>
> We thank the reviewer for raising the questions. The analysis on the noisy cases including both wrong labels and irrelevant variables has been added into the updated version (Section 6 and Appendix D, E, G). We studied the convergence properties of AND, OR, and XOR operators under training samples with random noise, categorized as wrong labels (Appendix D) and irrelevant input variables (Appendix E). A wrong label refers to an input that should be labeled as 1 but is instead labeled as 0, or vice versa. An irrelevant input variable, on the other hand, is one that does not contribute to the classification. We demonstrate that, with wrong labels, the TM  does not converge to the intended operators but can still learn efficiently. With irrelevant variables, the TM converges to the intended operators almost surely. Experimental results confirmed these findings (Appendix G).
>
> The citations have been properly updated in this version. We have also streamlined the text to reduce the overuse of commas, minimizing potential confusion for readers. Additionally, Section 2 and Appendix A have been revised to better explain the concept of TM and its components, which we believe will serve as a helpful and more convenient reference for readers.

---

> > ### Comment · Reviewer_M3eM · 2024-11-21
> > **Some clarifications and a short review of the revised version**
> >
> > I would like to thank the authors for their response and appreciate their efforts in revising the paper in accordance with the reviews.
> >
> > In the first part of their response, the authors suggest that publishing this paper at ICLR would enhance the visibility of Tsetlin Machines, which have close connections to deep neural models while offering appealing qualities such as transparency and interpretability. I want to clarify that I do not disagree with the authors regarding this aspect.
> >
> > However, it is important to note that this paper is not a survey on TMs promoting their practical interest; my review would have been entirely different if that were the case. The primary aim is to learn conjunctions or disjunctions in a transparent manner, using a Tsetlin Machine. As I pointed out in my review, a natural question arises:
> >
> > (Q) Are there existing transparent algorithms capable of learning interpretable models that express conjunctions or disjunctions under similar conditions?
> >
> > At the very least, the paper should include a detailed comparison with related work on interpretable machine learning. Since the goal is to learn simple concepts such as conjunctive or disjunctive clauses, a comparison with well-known results on concept learning is also crucial.
> >
> > In the revised version of the paper, the conditions under which learning operates remain ambiguous. From what I understand, the learning protocol is a supervised setting where examples are represented as $(x, y)$, with $x$ being a 2-dimensional Boolean vector and $y$ a Boolean label. According to Lines 159-162 and 234-236, it appears that the examples $(x, y)$ are independently drawn at random from a probability distribution $P$; however, this should be clarified. Additionally, it is assumed that all possible examples are included in the training set.
> >
> > In Sections 3 and 4, it is assumed that learning occurs under noise-free conditions. In this context, the goal for the TM model is to reach a fixed point (convergence) where the final state represents either a conjunction (as discussed in Section 3) or a disjunction (as discussed in Section 4). As indicated in my review, this scenario is very akin to the well-known "proper realizable PAC learning setting."
> >
> > To answer question (Q), the response is clearly yes when the target concept is one of the following: $x_1 \land x_2$, $x_1 \lor x_2$, or $x_1 \otimes x_2$. This can be achieved using, for example, the classic candidate elimination algorithm with 2-DNF as the hypothesis class. To clarify, 2-DNF represents the set of all disjunctions of conjunctions involving at most two literals. Therefore, the model produced as output is interpretable.
> >
> > In Section 6, alternative settings are examined where examples may be noisy or include irrelevant variables. First, it is important to clarify the concept of "noise," as there are various noise models in the machine learning literature (see, for example, Frénay and Verleysen, TNNLS ’14). Theorem 3 states that the TM does not converge, but is instead "recurrent." The term "recurrent" should be defined more clearly here.
> > On another note, it is known that conjunctions are properly PAC learnable under random noise when $ P $ is a product distribution (Mansour and Parnas, IPL ’88). Thus, the negative result presented in Theorem 3 should be compared with this positive finding.
> >
> > In the second setting, Theorem 4 focuses only on the naive case involving a single irrelevant variable. However, the issue typically explored in theoretical machine learning literature pertains to learning in the presence of many irrelevant variables. In such cases, examples take the form $ (x, y) $, where $ x $ is a $ d $-dimensional Boolean vector, and $ y $ is a Boolean label. If the target concept is a clause over $ k $ variables, there will be $ d - k $ irrelevant variables. It is noteworthy that the training set cannot include all possible examples unless exponential space is utilized. Thus, the question arises: Are TMs capable of learning in polynomial space clauses of length at most $ k $ when $ d - k $ irrelevant variables are present? In response, Valiant or Haussler’s papers offer elegant solutions to this problem, employing interpretable models and transparent algorithms with polynomial sample complexity.

---

> > > ### Author Response · Authors · 2024-11-23
> > > **Respond to the 2nd round of review**
> > >
> > > We would like to sincerely thank you for your professional, prompt, and insightful comments. Your feedback has inspired new findings and conclusions, and significantly enhancing the quality of the paper. We deeply appreciate your valuable contributions—thank you!
> > >
> > > In the revised paper, the main updates are marked in violet.
> > >
> > > Regarding the related work, it is no doubt that there are numerous existing methods that can learn conjunctions or disjunctions in an elegant way, and it is important to recognize and acknowledge the existing solutions and previous efforts. We therefore have added a paragraph in Section 1 summarizing several relevant studies with references. We understand that this addition is not exhaustive, but due to the constraints of the 10-page limit and the limited timeframe before the discussion phase concludes, we are unable to provide more rigorous theoretical or experimental comparisons in this submission. Within these constraints, we have chosen to focus on proving the capabilities of the TM. We recognize the need for an independent performance comparison paper, emphasizing existing studies within the domains of concept learning and PAC research. For future research, we plan to evaluate and compare their performance using realistic datasets, such as IMDB and MNIST, which offers an exciting and valuable direction for further exploration.
> > >
> > > Regarding your comment in the paragraph: "In the revised version of the paper, the conditions under which learning operates remain ambiguous. From what I understand......."
> > >
> > > Yes, your understanding is correct. It is supervised learning for 2-D Boolean vectors with a label. The samples are independently drawn at random from the distribution and all possible examples appear in the training set. We will make it clear in the paper.
> > >
> > > Regarding your comments in the paragraph: "In Sections 3 and 4, it is assumed that learning occurs under noise-free conditions...." and the paragraph: "To answer question (Q), the response is clearly yes when the target concept is one of the following: x1∧x2, x1∨x2, or x1⊗x2...".
> > >
> > > Yes, it is indeed akin to the well-known "proper realizable PAC learning setting", and numerous existing interpretable algorithms can learn it. However, the existence of other methods does not necessarily conclude that the TM is able to converge to the operators. Whether TM can also learn it or not was open and this paper gives an answer. Using this analysis on these toy-like operators, we can not only show convergence of TM for those cases, but also give insights in the operational concept of TM, helping researchers with better sense in hyper-parameter tuning in their applications of TM.
> > >
> > > Regarding your comment in the paragraph: "In Section 6, alternative settings are examined where examples.....".
> > >
> > > Thank you for your guidance regarding the noise type. It belongs to NCAR noise type defined in the paper (Frénay and Verleysen, TNNLS ’14), i.e., the occurrence of a wrong label is independent of the other random variables, including the true class itself.  Now we have clarified it.
> > >
> > > The “recurrent” system is the same as the one defined in a stochastic process. It is a process that revisits a particular state or set of states repeatedly over time with non-zero probability.  The recurrent property of TM indicates that there is a non-zero probability that it cannot learn the intended operator when the learning process terminates (i.e., once learnt, it may get out of the learnt state). However, it does not mean that the conjunctions are not learnable by TM at all. Instead, it is just not with probability 1.
> > >
> > > After studying the definition of PAC learnable and the suggested paper, we found that our conclusion in Theorem 3 does not conflict with the conclusion that conjunctions is PAC learnable. (We cannot find the paper in IPL in the year 1988, but we found “Mansour and Parnas, learning conjunctions with noise under product distributions, IPL 98”. I hope this is the paper that you meant).  In the definition of PAC learnable, we find that the learning probability is $1-\delta$, which allows $\delta$ as a room/margin for not learning the intended operators. Here the probability that TM cannot learn the intended operator can be the $\delta$ in PAC.
> > >
> > > Although the TM does not converge to any point with probability 1, from numerical experiments, we have observed that the TM can indeed learn the intended operator with high probability. For example, we have observed from experiments that TM can learn the intended operator for all experiments for the configurations in Appendix G1.  All in all, these two concepts do not collide.  We have provided a remark on this aspect in the updated version.

---

> > > ### Author Response · Authors · 2024-11-23
> > > **Respond to the 2nd round of review: Continue**
> > >
> > > Regarding your comment in the paragraph: "In the second setting, Theorem 4 focuses only on the naive case..."
> > >
> > > Many thanks for your inspiration. After reviewing the proof of irrelevant variables, we realized that the proof does not constraint to 1 bit of irrelevant input. In fact, we can extend it to multiple bits of irrelevant inputs, and in theory, one absorbing state still exists with infinite number of irrelevant input variables, because it is possible for TM to exclude all irrelevant input or its negations (with infinite times and samples).  The statement and the proof are updated.
> > >
> > > Regarding polynomial sample complexity, although not yet theoretically proven, experimental results and our intuition strongly suggest a positive answer to this question.
> > >
> > > In our experiment (the very end of the appendix), we configured the TM to learn a noisy XOR function with 2 useful input bits and 18 irrelevant input bits (d=18, k=2). Remarkably, the TM was still able to learn the XOR operator with 100% accuracy using just 5000 samples. If all possible input combinations were required in the training samples, we would require 2^20=1048576 samples. Clearly, the TM does not rely on the entire combinatorial input space to learn effectively. For this experiment, we used the following hyperparameters:  N=128, m=20, T=10, s=3, label noise 0.1.
> > >
> > > Intuitively, a polynomial space for training samples seems reasonable. This is because the TM can independently update the actions of a TA within a clause, as long as the clause value and the literal value are determined by the training sample. In other words, once the clause value and the literal value are known, the transitions triggered by Type I and Type II feedback are fully determined. As a result, the TM does not need to observe all possible combinations of irrelevant inputs to learn. Instead, as long as the statistical irrelevance of certain inputs is demonstrated in the training samples, the corresponding TA transitions will be triggered accordingly. This enables the TM to learn without requiring exhaustive coverage of the input space. In addition, the initial states of all TAs are on the “exclude” side and the TAs are trained by samples to include a certain literal, helping with excluding irrelevant variables.

---

### Official Review · Reviewer_EsZt · 2024-11-01

**Soundness:** 2
**Presentation:** 2
**Contribution:** 1
**Rating:** 3
**Confidence:** 2

**Summary:**

The paper aims to show the convergence of Tsetlin machines (TM), a relatively new machine learning approach based on Tsetlin automata (TA), for AND and OR operators. After previous work has done work on the convergence for the XOR operator, as well als the IDENTITY and NOT operators, this fills the gap to show convergence for standard binary operators.

**Strengths:**

The problem discussed in the paper is a natural continuation of previous works and, based on this line of research, seems to be of interest for the community. The relevance for the paper is hinted at by citing several use cases for TM.

The structure of the paper is straight forward, with each of the two main sections being dedicated to one of the two operators for which convergence is discussed.

I like the loop-back to the previous work on the XOR-operator, establishing a more detailed comparison to other recent results.

**Weaknesses:**

From this paper, it is very hard to grasp the concept of how TM are used for machine learning applications for anyone not familiar with TM. The usefulness of the main theoretical results of the paper -- the convergence of TM for the AND and OR operator -- also very much remains unclear since it is not discussed.

Also on a technical level this paper does not convey how TM and TA work in the first place. Despite reading through section 2 and appendix A, I did not gain an understanding of what a TM is. To exemplify this, even the start of the first sentence in the preliminaries ("A TM that is to learn the sub-patters of class $i$ is formed by $m$ teams of TAs in the form of clauses, $C^i_j(X), j=1,2,...,m$, where $X$ is the input of a TM, and $X=[x_1,x_2,...,x_o],x_k\in ${$0,1$}$, k=1,2,...,o$.") raises several questions that are never addressed in the paper:

* What is a class here?
* What do you mean by sub-patterns of a class?
* How is a TA even defined?
* From what I read from related work, a TA is a probabilistic automaton. How is it represented by a clause?
* What is a team of TAs? Is it different to just a set of TAs (and if yes, in which way)?
* What are the $x_k$ in the input to a TM?

Many of the definitions, e.g. the definition of $\xi^i_j$ (line 90) or the concept of polarity (line 102ff), are not given in a formal way but only in plain language, making the theoretical foundation of the paper very imprecise.

Most importantly, "convergence" is neither defined nor explained for TM. This makes one of the main statements of the paper, Theorem 1, impossible to understand from the paper alone.

For future revisions I **highly** suggest to make the paper self-contained and make sure that anyone not familiar with TM (which I imagine will be a large fraction of audiences) is able to follow the ideas of the paper. I also want to mention that large parts of the preliminaries including figures, are copied from related work, such as Jiao et al. (2022), Zhang et al. (2022) and Granmo (2018). While they are properly cited and this only affects preliminaries (and thus I do not see concerns regarding plagiarism), I think it would be highly beneficial for this paper to set up its definitions, tailored to the setting of this paper, in order to enhance understandability.

Based on the lack of formal definitions I was not able to follow sections 3 and 4 and cannot give detailed feedback on it.

At the end of section 4, experiments validating the theoretical results are mentioned. However, discussion on the experiments are fully shifted into the appendix. In my opinion, they should also be discussed in the main part of the paper.

**Questions:**

Can you clarify the questions regarding the theoretical background I raised in the bullet points in the weaknesses section? Should these questions have been resolved from any part of your paper (if yes, where?), or do you assume these as a background a reader should have?

In Eq (1), what is the semantic difference between the testing and training formula? The usual convention is for the empty conjunction to be defined as true, if you differ from that this should be mentioned.

Can you give an example for a machine learning task where TM would be applicable? What would be the TM input and output in that example and how would the TA within the TM look like?

How do the theoretical results you show in this paper affect the use of TM in practice?

minor comments:

* the citation style without parentheses is strenuous to read, please use \citep (see point 4 in ICLR style guidelines)
* line 56: "...while this paper searches for absorbing states" -- what does "this paper" refer to in this sentence? From the grammar I would assume your own submission, but form context rather the paper by Zhang et al. (2022) or Jiao et al. (2022)
* line 64 "learnt" -> learned

---

> ### Author Response · Authors · 2024-11-20
>
> Thank you for your questions and valuable feedback.
>
> We have updated Section 2. Here we use TM learning the OR logic as an example. A sample is classified into the OR class if its two feature bits and label follow the OR logic: 01=>1, 10=>1, 11=>1, 00=>0. Note that once the TM learns the pattern of the features that produces an output of 1, it inherently learns the complementary pattern that produces 0. Hence the TM's learning and reasoning can be understood primarily as identifying the pattern that results in an output of 1.
>
> In the OR logic, 01, 10, 11 are sub-patterns that produce an output of 1, and can be represented by clauses $\neg x_1 \wedge x_2$, $x_1 \wedge \neg x_2$, and $x_1 \wedge x_2$, respectively.
>
> A clause is learned by a TA team, and a TM can be composed of multiple TA teams. A TA team is a set of TAs, each responsible for handling one literal. A literal is an input feature: $x_1$ or $x_2$, or the negation: $\neg x_1$ or $\neg x_2$. In our example, each TA team consists of four TAs, i.e., two TA pairs, managing four literals: $x_1, ~\neg x_1, ~x_2, ~\neg x_2$, respectively.
>
> A TA decides whether to Include or Exclude its literal by its current state, which changes according to the feedback. In our example, if TA1 includes $x_1$, TA2 excludes $\neg x_1$, TA3 excludes $x_2$, and TA4 includes $\neg x_2$, the resulting clause is $x_1 \wedge \neg x_2$. An example how the feedback controls the TA state transition can be found in Section 3.1 (scenarios 2 & 5 ) of paper: Zhang, et al. "On the Convergence of Tsetlin Machines for the IDENTITY- and NOT Operators." TPAMI2022.
>
> A TM learns the pattern of the OR relationship from the input samples that follow the OR logic (training). As a result, some TA teams converge to clauses like $\neg x_1 \wedge x_2$, others to $x_1 \wedge \neg x_2$, $x_1 \wedge x_2$, $x_1$ or $x_2$, all outputting 1. The process of determining whether an input $[x_1, x_2]$ conforms to the OR logic involves summing the outputs of all the clauses. Let's assume we have three TA-teams, each converging to one of the sub-patterns, then the sum could be $sum=(\neg x_1 \wedge x_2)+x_1+x_2$. If a test sample: {$[x_1, x_2], y$}={[0,1], 1} is fed to the TM, the output would be $sum=(1 \wedge 1)+0+1=1+0+1=2$, indicating two TA teams vote for positive classification. If a threshold  $Th$ is defined as 1, as $sum>Th$, TM confirms the sample following the OR logic (testing).
>
> In general, the input to a TM is denoted as $X=[x_1,x_2,...,x_o]$, where $x_k$ takes a boolean value, i.e., $x_k \in$ {0, 1}, and k=1,2,...,o. In the above example, o=2, and $X=[x_1, x_2]$.
>
> I hope the example clarifies the workings of the TM, its components, the relationship between a clause and a TA team, and how a clause is constructed based on the decisions of TAs to include or exclude literals.
>
> The definition of $\xi^i_j$ and the concept of polarity are provided in plain language rather than formally, as they are not directly involved in the proof of convergence. However, we added more details in Section 2 and Appendix A to clarify the concept, addressing the lack of clarity in the initial version.
>
> We define a TM as having converged when the states of its TAs no longer change. This definition has been added in the updated version.
>
> Empty clauses have been defined for training and testing in the updated version.
>
> Besides the example provided above, we also list several research papers, where you can find detailed information on how the input is organized and how TMs are applied to more complex problems.
> - Yadav, R., et al. "Human-level interpretable learning for aspect-based sentiment analysis." AAAI2021.
> - Yadav, R., et al. "Robust interpretable text classification against spurious correlations using AND-rules with negation." IJCAI2022.
> - Seraj, R., et al.  "Tsetlin machine for solving contextual bandit problems." Neurips2022.
> - Maheshwari, S. et al., "REDRESS: Generating Compressed Models for Edge Inference Using Tsetlin Machines," TPAMI2023
>
> TM is also utilized in industrial and public sector applications:
> - Using Tsetlin machine for GPS jamming detection by NKOM (Norwegian communication authority): https://nkom.no/aktuelt/nkom-tester-norsk-ki-modell-i-jakten-pa-gps-forstyrrelser
> - Developing sustainable AI by the company Literal Lab: https://www.forbes.com/sites/charlestowersclark/2024/09/20/unlocking-sustainable-ai-the-game-changing-tsetlin-machine-approach/
>
> TM is a symbolic AI algorithm with the potential to be organized into a deep connection structure similar to neural networks. This is a novel algorithm design that is worth studying and exploring. The convergence of TM on the basic logic concept helps fundamentally understand the design and demonstrate its effectiveness in basic classification problems, thereby enabling its expansion to more complex application scenarios. In addition, uncovering the functionalities of hyperparameters such as s, T, and m offers valuable insights and practical guidance.

---

> > ### Comment · Reviewer_EsZt · 2024-11-22
> >
> > First, I acknowledge the response and want to thank the authors for their thorough response, additional explanations, and already incorporating the suggested changes into the paper.
> >
> > In another response, you write
> >
> > > We believe that publishing at prestigious conferences like ICLR will bring greater visibility to TM
> >
> > and in this response
> >
> > > TM is a symbolic AI algorithm with the potential to be organized into a deep connection structure similar to neural networks. This is a novel algorithm design that is worth studying and exploring.
> >
> > and (seeing the related work on, and applications of TM) I agree with both of those statements. However, to encourage a discussion on TM and bring it to an audience like ICLR (where most people will not be familiar with TM) a solid foundation of TM must be established within the paper. In other words, the paper must be mostly self-contained. As a parallel example, almost all reinforcement learning papers precisely define MDP in their preliminaries (instead of only giving a textbook reference), depsite MDP being far more wide spread than TM.
> >
> > The updated version, while somewhat clearer, still suffers from similar issues:
> >
> > * l. 72: "The TA is a 2-action learning automaton..." - what kind of automation? DFA/NFA, or over infinite words...? What is the state space, initial states and transitions function?
> > * "...whose job is to decide whether to Include/Exclude its literal in/from the clause, and the decision is determined by the current state of the TA." - how does it decide on exclusion/inclusion exactly? Which is "the clause" in this context - Is each TA assigned exactly one clause? How do TAs "vote"?
> > * l.158 "A TM has converged when the states of its TAs do not change any longer." - I assume you mean the current state, and not the state space since the TA are fixed, right? Still, this does not clarify anything since, again, TAs have not been formally defined in the preliminaries.
> >
> > Additionally, for a clear presentation, it is similarly crucial to limit the information to the necessary bits, especially in view of space constraints. As such, if you say $\xi^i_j$ and polarity are not necessary to show convergence of TMs, then why include them in the paper? To me, this only adds unnecessary questios for readers unfamiliar with TM (which, again, will probably be most people at ICLR) and distracts from the main contribution of the paper. Again, I think this is an artifact of simply cpoying the preliminaries from related work.
> >
> > While I appreciate the effort towards addressing my concerns, my main critique point still stands and even in the updated version I think this paper is completely inaccessible for readers unfamiliar with TMs. As such I will keep my score.
> >
> > For future revisions I once again emphasite my suggestion to fundamentally rework the preliminary section such that can be understood by anyone with a formal background, even outside of TMs.

---

> > > ### Author Response · Authors · 2024-11-25
> > >
> > > Many thanks for your detailed comments.
> > >
> > > We understand the reviewer’s concern regarding the presentation, and we agree that ensuring the paper is as self-contained as possible within the main body is ideal. However, the 10-page space limit restricts our ability to provide a comprehensive explanation of the TM's operational concept along with its proofs. As a result, we have prioritized presenting the most critical notations and aspects of the TM in the main context. This inevitably assumes prior knowledge of learning automata, propositional logic, and preferably TM itself.
> > >
> > > In the proof, although the variable $\xi$ is not directly utilized, we decide to include it to illustrate how a clause is constructed. Although we couldn’t elaborate further within the main paper, we believe the provided references and appendix should be sufficient for an AI specialist to understand the underlying concepts. The fact that three out of four reviewers in this process have provided insightful comments suggests that, while the paper may require some additional reading of the appendix and references, it is not entirely inaccessible.
> > >
> > > On another note, the Tsetlin Automaton (TA) is a type of Fixed Structure Stochastic Automaton (FSSA) with a finite number of states. Once the TM converges, the transitions among the states in the TAs cease.
> > >
> > > Finally, based on your questions, we suspect there may be some confusion between learning automata and the automata used in automata theory (e.g., DFA or NFA). While they share similarities, they are fundamentally different. Learning automata determine their actions based on past interactions with their environment, aligning with reinforcement learning principles. In contrast, automata in automata theory are self-operating systems designed to follow a predefined sequence of operations or respond to predetermined instructions.

---

### Official Review · Reviewer_dUop · 2024-11-02

**Soundness:** 4
**Presentation:** 3
**Contribution:** 4
**Rating:** 8
**Confidence:** 3

**Summary:**

The paper's main contribution addresses the missing piece for TM's convergence: AND and OR operators. Previous studies have analyzed (and published after peer review) the 1-bit NOT and IDENTITY operators and the 2-bit XOR case. This paper fills the remaining gap (as far as I understand) in 2-bit AND and OR operators. The final result of this paper is to complete the bigger picture of logical operators interpreted over the Boolean algebra. The paper is already in a solid position to be accepted, but minor changes could improve the overall comprehensibility.

**Strengths:**

- **S1:** Clear mathematical formulation.
- **S2:** Precise collocation in the relevant literature, and therefore, the paper addresses a novel problem
- **S3:** Great illustrations to help the reader.

**Weaknesses:**

- **W1:** As far as I understand, the results are limited to 2-bit cases. This limitation is essential as real-world data rarely have only 2-bit input.
- **W2:** I may be wrong, but perhaps the interdependence between clauses is the reason previous proofs and those presented in this paper have not generalized to >2-bit cases.
- **W3:** To learn TMs, non-binary datasets must necessarily be transformed into binary ones. As far as I understand the literature, further research is needed to address this issue. Applying naive techniques to binarize datasets could potentially introduce biases during the transformation.
- **W4:** After a quick search on the Internet, I found an arXiv version of this paper with the same name and typos (see https://arxiv.org/abs/2109.09488). For example, search for ".)." (i.e., extra ".") in both PDFs and there will be a perfect match for the same typos (in the four cases). As such, the authors are likely the same. I am not sure if that was the original intention, but nevertheless, this strategy does not entirely fulfill the double-blind process.

**Questions:**

- **Q1:** What are the challenges of generalizing this result and the previous ones for XOR, AND, and OR to the >2-bit cases?
- **Q2:** What are the challenges when having numerical data as inputs? How do you suggest handling non-binary datasets in TMs? Specifically, what techniques could binarize data while minimizing potential biases?
- **Q3:** Have you considered generalizing beyond propositional logic? I am fond of logic, and it would be nice to know if the authors have in their research agenda to extend their results to more expressive logics, such as modal and/or first-order logic. This aspect is also essential as a considerable amount of real-world data comes from signals hardly modeled using static Boolean vectors/masks.
- **Q4:** I don't understand why the authors claimed in Theorem 2 that "[...] the OR logic [...]" when their result applies to 2-bit inputs only. Indeed, Equation 4 (right before Theorem 2) presents four 2-bit inputs. Am I missing something here? Consider clarifying in Theorem 2 that the result applies specifically to 2-bit OR logic, or explain how it generalizes to n-bit cases if that's the case.
- **Q5 (minor):** Are there any reasons why you have used a poorly formatted citation style throughout the paper? I would have expected to see, e.g., (Granmo, 2018) instead of Granmo (2018).

---

> ### Author Response · Authors · 2024-11-20
>
> Thank you for recognizing the value of this paper and for providing valuable suggestions for improvement.
>
> W1: This is correct. This paper analyzes the learning process of the TM on the 2-bit AND and OR logic. As pointed out in W2, generalizing the proof to more than 2 bits is challenging. However, in the updated version, we added the analysis on the noisy case, including wrong labels and random irrelevant input variables.
>
> W2/Q1: This is correct. Clauses are interdependent, so are the literals within a single clause. Due to the interdependency, as the number of bits increases, the state space grows larger, making the analysis significantly more challenging. A paper attempting to provide a generalized proof (not on TM, but a simpler algorithm than TM) can be referred to https://arxiv.org/abs/2310.02005. In that work, the feedback strategy was simplified to decompose the dependency, resulting in slower learning as a consequence.
>
> W3/Q2: This is an excellent question and indeed one of the most challenging aspects of applying TM. In the application of Regression TM, simple thresholding was used to convert continuous inputs into binary representations (Abeyrathna D. K., et al, "The regression Tsetlin machine: a novel approach to interpretable nonlinear regression." Phil. Trans. R. Soc. A 378: 20190165. http://dx.doi.org/10.1098/rsta.2019.0165). In image processing, adaptive Gaussian thresholding was applied to binarize images (Sharma, J., et al. "Drop clause: Enhancing performance, robustness and pattern recognition capabilities of the tsetlin machine." AAAI2023). As the reviewer pointed out, the process of booleanization in real-world data can introduce varying degrees of bias. There is ongoing research in this area. The latest progress can be found in the following paper (https://arxiv.org/pdf/2406.02648), which explores using hyper-vectors to represent inputs.
>
> W4: a complete coincidence...
>
> Q3: We have not yet explored more expressive logics, such as modal and first-order logic. However, this is an intriguing idea, and we will evaluate its feasibility and consider incorporating it into our future research agenda. Thank you for your valuable suggestion. There is ongoing research on the so-called Graph TM, where the idea may be somewhat similar to the suggestion here.
>
> Q4: Thanks for the observation. You are right, it is 2-bit OR. Theorem 2 has been updated.
>
> Q5: The citation style has been updated.

---

> > ### Comment · Reviewer_dUop · 2024-11-21
> >
> > I acknowledge reading the rebuttal and thank the authors for their answers.
> >
> > I respect the perspectives of the other reviewers and agree that improving the presentation for a broader ICLR audience is crucial. However, I believe it's essential to recognize that while ICLR has traditionally focused on deep learning, we should not overlook different strategies for learning representations, such as the potential contributions of symbolic AI.
> >
> > I have no further questions or observations at this time.

---

### Official Review · Reviewer_japi · 2024-11-03

**Soundness:** 4
**Presentation:** 3
**Contribution:** 4
**Rating:** 8
**Confidence:** 2

**Summary:**

This paper studies Tsetlin Machines (TMs). It builds on earlier papers, and in particular on an earlier paper that shows that clauses will converge almost certainly to the XOR logic operator. In this paper, the author(s) intend to answer the same question for the AND and OR operators.
The author(s) notes that upon completing this project, the convergence of all basic operators in Boolean algebra will be established. This certainly makes the questions relevant, and the paper does indeed address these questions.
The author(s) gives proofs for their claims, both in detail in the appendix, as well as the idea in the main text (wherever appropriate). Because of this, and because as far as I can tell the proofs seem correct, their claims are well supported.
The new knowledge that this paper reports is certainly interesting for anyone who studies or uses TMs. Therefore, this paper is relevant for the ICLR community.

**Strengths:**

This paper has a very clear project, which I believe is important, and relevant to the ICLR community. The paper answers their two questions, namely: "do TMs converge on the AND and OR operator?" positively, by giving proofs of these claims. The proofs seem correct, but I was unable to check all the details.
I appreciated the choice of the author(s) to include "proof ideas" in Sections 3 and 4.
The writing and presentation is mostly very good. I have some slight suggestions for improvement, which I will list in the section "Weaknesses".

**Weaknesses:**

I think this is a good paper, so my weaknesses are less important than the strengths I identified in the section 'Strengths".

The main weakness, in my opinion, is that the paper does not convey the idea behind a TM well. What are the main ideas behind a TM? What are they used for? Being a non-expert, I had to look this up.
I understand and appreciate that it might not be possible to accommodate for this in general due to page limitations, but I do recommend that the author(s) streamlines some of the presentation, in particular:
- On line 052, the author(s) talks about the hyperparameter T. A bit later, on line 105 they talk about a threshold Th, which later turns out to be related to T. Later on, they mostly talk about T again, and relate this twice to Th. It is still not clear to me how to think about Th and T exactly. For instance in Eqs (2) and (3) it would be useful to know what T is. I recommend to make this more clear.
- On Lines 139 – 147 the authors talk about the two types of feedback they generate. It would be helpful to know an (informal) argument about why these feedbacks precisely are used.

I end this section with a list or minor remarks and typos:
- Line 172: It would be useful to indicate the start and end of the proof idea. As I said earlier, I do like the choice of including the proof idea in the main text.
- Line 226: "we can concluded that" shoud be "we can conclude that".
- Line 228: "we must frozen TA3" shoud be "we must freeze TA3" or something along those lines.
- Line 290: "samples following Eq. (8) is given' should be "samples following Eq. (8) are given'.
- Line 291: "the absorbing states for Eq. (t) disappears" should be "the absorbing states for Eq. (t) disappear".
- Line 452: I believe that "We conjuncture" should read "We conjecture".

**Questions:**

A question I wonder about is this. It seems to me that the analysis for the OR statement is much harder than that for the AND statement.
Since the convergence for the XOR statement is already established earlier, and for the AND statement in Section 3 of the current paper, could this be used to derive the convergence for the OR statement using a OR b iff (a XOR b) XOR (a AND b), in a much quicker fashion? Perhaps there are technicalities that prevent such an inference, but it would be interesting to know which they are, and if they could be circumvented.
The currect Section 4 gives us more insight into the hyperparameter T, so I do not suggest to replace this section. I am merely interested in whether such inferences are possible.

---

> ### Author Response · Authors · 2024-11-20
>
> We sincerely thank the reviewer for recognizing the value of this paper and for providing constructive suggestions.
>
> In the updated version, we have revised Section 2 to better introduce the workings of the TM, its components, the relationship between a clause and a TA team, and how a clause is constructed based on the decisions of TAs to include or exclude literals. Clearer explanations and usage of the hyperparameter T and the threshold Th has also been added. The hyperparameter T is used to regulate the clauses allocation in training, to avoid the situation that a majority of the TA teams learn only a subset of sub-patterns, forming an incomplete representation. The threshold Th is used in testing, if the voting sum of all the clauses surpasses Th, the input sample will be identified as belonging to the class. Additionally, specific references have been added to help readers more easily find examples that demonstrate how feedback influences the state transitions of TAs. Spelling errors have also been corrected in the new version, and the start and end of the proof idea have been marked as well.
>
> Regarding the Question: We fully agree that we can construct OR from the other operators. This also made the authors of the paper believe, at the beginning, that once the convergence of XOR and AND have been proven, the convergence of OR can be derived and hence trivial. However, this is not the case. The fact is that the convergence of TM on different logic operators needs separate investigations.
>
> The reasons are two-folds. Firstly, the TM, by nature, learns the operators in a direct manner instead of a derived manner. For example, A properly trained TM represents the two sub-patterns in the XOR logic using two distinct clauses: $\neg x_1 \wedge x_2$ for $0 1$ and $x_1 \wedge \neg x_2$ for $1 0$. It does not rely on clauses representing complex combinations of AND, OR, and NOT to derive XOR indirectly. Secondly, as already pointed out by the reviewer, although TM can converge almost surely to these operators, the approaches for the TM to converge to different operators are different. Indeed, the proofs on the convergence of different operators reflect various facets of TM in its learning mechanism. Specifically, the proof of XOR reveals how a TM learns mutually-exclusive sub-patterns and what the functionality of T is in the process of learning. While in the proof of OR operator, it shows how the non-mutually-exclusive sub-patterns (01=>1, 10=>1, 11=>1) can be learned jointly by the TM, and the clauses can be $\neg x_1 \wedge x_2$, $x_1 \wedge \neg x_2$, $x_1 \wedge x_2$, $x_1$, and $x_2$, with the latter two being the combinations of two of the former clauses.

---

> > ### Comment · Reviewer_japi · 2024-11-25
> > **Reply to the authors**
> >
> > I'd like to thank the authors for their reply, and improvements to their paper. I believe that their Section 2 is more clear now. I also appreciate the authors's reasoning about my question regarding the connection between convergence for AND, OR and XOR operators.
> > Therefore, I still believe this is a good paper.

---

### Meta-Review · Area_Chair_x8Gx · 2024-12-22

**Metareview:**

The paper proves that Tsetlin Machines can learn OR and AND operators.  This is a theoretical paper that advances the understanding of the capabilities of Tsetlin Machines.  The strengths of this paper are the theoretical contributions.  The weaknesses include the presentation that assumes expertise with Tsetlin Machines and a lack of discussion and theoretical comparison to other work in learning theory that shows that other models can also learn the OR and AND operators.

**Additional Comments On Reviewer Discussion:**

Discussion among the reviewers focused on the revisions to the paper to improve its accessibility.  The revisions to Section 2 and Appendix A helped a lot since it is now possible for a reader unfamiliar with Tsetlin Machines to grasp what they are.  However the descriptions are still difficult to understand and require multiple reads including Appendix A.  The area chair personally red the paper and shares the reviewers' concern about the accessibility of the paper.  While the paper now provides suitable definitions for most concepts, there is still no description of how Tsetlin Machines are updated when learning a concept such as the AND and OR operators.  Appendix A says "After Type I Feedback or Type II Feedback is generated for a clause, each individual TA within each clause is given reward/penalty/inaction according to the probability defined, and then the states of the corresponding TAs are updated."  However there is no description of the update.  This is problematic since the paper proves that Tsetlin machines can learn the AND and OR operators, but there is no description of the learning procedure that the proofs refer to.  That being said, the learning procedure is described in other papers, but there should be at least a short outline for the reader to appreciate the proofs.

The revised paper includes in the introduction some references to previous work in learning theory regarding the AND and OR operators.  However, a more in depth discussion is needed.  In fact previous work shows that it is possible to learn AND and OR operators in polynomial time and finite sample bounds are provided.  A more in-depth discussion is needed to position Tsetlin machines with respect to existing work in learning theory.

---

### Decision · Program_Chairs · 2025-01-22

Reject